# Demonstration-Guided Continual Reinforcement Learning in Dynamic Environments

## Abstract

Reinforcement learning (RL) excels in various applications but struggles in dynamic environments where the underlying Markov decision process evolves. Continual reinforcement learning (CRL) enables RL agents to continually learn and adapt to new tasks, but balancing stability (preserving prior knowledge) and plasticity (acquiring new knowledge) remains challenging. Existing methods primarily address the stability-plasticity dilemma through mechanisms where past knowledge influences optimization but rarely affects the agent's behavior directly, which may hinder effective knowledge reuse and efficient learning. In contrast, we propose demonstration-guided continual reinforcement learning (DGCRL), which stores prior knowledge in an external, self-evolving demonstration repository that directly guides RL exploration and adaptation. For each task, the agent dynamically selects the most relevant demonstration and follows a curriculum-based strategy to accelerate learning, gradually shifting from demonstration-guided exploration to fully self-exploration. Extensive experiments on 2D navigation and MuJoCo locomotion tasks demonstrate its superior average performance, enhanced knowledge transfer, mitigation of forgetting, and training efficiency. The additional sensitivity analysis and ablation study further validate its effectiveness.

## 1 Introduction

Reinforcement learning (RL) has shown significant potential across diverse domains, including video games Mnih et al. (2013), robotics Parisi et al. (2019), drone navigation Kaufmann et al. (2023), and energy management Lu et al. (2022; 2023). The dominant paradigm in RL typically focuses on optimizing performance for a single task where the underlying Markov decision processes (MDP) remain static over time. However, this assumption is often unrealistic as any aspect of the MDP may change in practice. In such non-stationary environments, RL agents frequently suffer from catastrophic forgetting, where newly acquired knowledge interferes with previously learned behaviors.

To make RL more applicable in realistic settings, it is necessary to equip RL agents with the ability to continually learn and adapt to evolving environments, much like human learning. This is referred to as continual reinforcement learning (CRL) Hadsell et al. (2020); Khetarpal et al. (2022); Wolczyk et al. (2022). A central challenge is achieving a balance between retaining old knowledge (stability) and acquiring new knowledge (plasticity), known as the stability-plasticity dilemma.

Existing methods, including replay-based Rebuffi et al. (2017); Ramapuram et al. (2020); Lopez-Paz & Ranzato (2017), regularization-based Li & Hoiem (2017); Kirkpatrick et al. (2017); Zenke et al. (2017); Nguyen et al. (2017), and parameter isolation-based approaches Rusu et al. (2016); Xu et al. (2021); Fernando et al. (2017), focus on addressing the stability-plasticity dilemma within neural networks (NNs), where the knowledge from past experiences mainly influences optimization during training but rarely affects the agent's behavior directly. For example, replay methods store past transtions in a replay buffer (often mixed with online data) and replay then when learning new tasks. However, prior knowledge is encoded in model parameters and does not explicitly specify the actions the agent should take during interaction with the environment. As a result, its influence on the agent's behavior is indirect. Despite NNs being powerful

function approximators Hornik et al. (1989), their black-box nature complicates the separation of knowledge retention from policy learning, which may limit the effective knowledge reuse and efficient learning across tasks. We refer to this challenge as the knowledge decoupling problem.

In addition, many methods necessitate modeling the environment by representing it as a set of factors to facilitate fine-tuning of previously learned policies. It requires parameterizing the current task or assessing task similarity to identify relevant context-specific knowledge for transfer Nagabandi et al. (2018); Wang et al. (2021); Zhang et al. (2023). However, rigid environment modeling can increase computational costs and introduce additional uncertainty, which we refer to as the environment modeling problem.

To address the aforementioned issues, we explore CRL from a different perspective. Human learners often rely on external sources such as reference materials or expert demonstrations that can be repeatedly consulted to directly shape the behavior in different situations. This motivates us to represent knowledge using reusable, external demonstrations to provide direct guidance on the agent's exploration behavior and accelerate efficient adaptation across tasks, particularly in cases where explicit environment modeling is unnecessary or infeasible.

We propose a novel CRL framework, i.e., *demonstration-guided continual reinforcement learning* (DGCRL), which stores prior knowledge in the form of demonstrations into an external, self-evolving demonstration repository. The demonstrations can be derived from trajectories, pre-trained policies, or human experts, even if they are sub-optimal. In this study, we use trajectory-level action sequences as demonstrations. They serve as the guide policy, providing a curriculum of starting states for the exploration policy, directly guiding the RL exploration behavior and promoting faster learning and adaptation across tasks. Compared with other continual learning methods, where prior knowledge mainly influences the agent *indirectly* by providing additional gradient signals during parameter updates, these approaches do not alter the agent's behavior during interaction with the environment. The rollout policy remains fully determined by the current exploration policy, and thus the state visitation distribution is unchanged. In contrast, demonstrations in DGCRL are executed during part of each rollout, allowing the guide policy to *directly* determine the agent's actions and state visitation distribution. This direct behavioral guidance enables DGCRL to place the agent in informative regions of the state space and significantly accelerate adaptation under environment changes.

For each new task, DGCRL selects the most relevant demonstration and gradually transitions from demonstration-guided exploration to fully self-exploration through a curriculum schedule, where the influence of demonstrations is gradually reduced. To achieve this, we extend the jump-start reinforcement learning (JSRL) Uchendu et al. (2023) method from single-task to continual learning settings by dynamically reselecting the most relevant demonstration for each new task, and incorporating a self-evolving demonstration mechanism Lu et al. (2024): if a newly learned policy outperforms existing demonstrations, we store it as a new demonstration in the repository to ensure ongoing knowledge retention and improvement.

DGCRL is capable of being integrated with various RL algorithms; in this study, we adopt an actor-critic algorithm, i.e., twin delayed deep deterministic policy gradient (TD3) Fujimoto et al. (2018), due to its proven advantages in continuous control tasks. Extensive experiments have been conducted on 2D navigation and MuJoCo locomotion tasks, where the transition and/or reward functions change across tasks, to study the properties and validate the effectiveness of DGCRL.

The contributions of this work are summarized as follows:

1. **Storing prior knowledge as external demonstrations.** We investigate the issues of existing CRL methods and propose DGCRL, a framework that stores prior knowledge as external demonstrations. DGCRL leverages demonstrations to directly guide the agent's exploration and adaptation, enabling effective knowledge reuse and efficient learning without the need for environment modeling.

2. **Dynamic curriculum-based exploration.** We introduce a dynamic curriculum-based strategy where the agent dynamically selects the most relevant demonstration for each new task. The demonstration can provide direct guidance on RL exploration and adaptation via a curriculum, allowing the agent to gradually transition from demonstration-guided exploration to fully self-exploration. Meanwhile, a self-evolving mechanism is incorporated to ensure the demonstration repository continuously improves during training.

3. **Empirical validation on 2D navigation and Mujoco locomotion.** Extensive experiments on robotics control tasks show that DGCRL significantly outperforms baselines in terms of average performance, forward transfer, mitigation of forgetting, and training efficiency. We also highlight the limitations of the conventional forgetting metric. The sensitivity analyses shows the impact of varying demonstration quantities on the performance. Furthermore, the ablation studies underscore the importance of resetting both the actor and critic components for optimal performance, and demonstrate the advantages of combining self-evolving demonstration guidance with curriculum learning in DGCRL.

We outline the related work in Section 2 and introduce the problem formulation in Section 3. The details of the DGCRL method is presented in Section 4. The experiment settings and results are shown in Section 5 and Section 6. We finally conclude this study in Section 7. In addition, the implementation details are provided in A, and a theoretical analysis showing the upper bound on regret and sample complexity is provided in B. The source code of DGCRL is available on GitHub [1].

## 2    Related Work

### 2.1    Continual Learning

Continual Learning has been extensively studied in the supervised learning domain and has recently received increasing attention in RL due to its implications in autonomous learning agents and robots Bagus et al. (2022); Parisi et al. (2019). It refers to how artificial systems learn incrementally from continuous streams of information in non-stationary environments, without the need to retrain from scratch. A set of related but distinct tasks need to be completed sequentially, where acquiring new knowledge often leads to catastrophic forgetting of previously learned tasks Hadsell et al. (2020). Limited resources such as computational budget and storage capacity should also be considered.

A variety of approaches have been investigated in this area, often categorized into three types: replay methods, regularization-based methods, and parameter isolation methods, depending on how task-specific information is stored and used throughout the continual learning process De Lange et al. (2021). With replay methods, previous task samples including raw samples or generated pseudo-samples are stored and relayed when learning new tasks, either as model inputs for rehearsal Rebuffi et al. (2017); Ramapuram et al. (2020) or used for constrained optimization of new tasks Lopez-Paz & Ranzato (2017). Regularization-based methods add an extra term to the loss function to regularize model parameters, using techniques such as knowledge distillation to provide an auxiliary target for the network being trained Li & Hoiem (2017), or by penalizing significant updates to important parameters during new task learning Kirkpatrick et al. (2017); Zenke et al. (2017); Nguyen et al. (2017). Parameter isolation methods protect model parameters for past tasks through dynamic architecture such as progressive neural networks Rusu et al. (2016); Xu et al. (2021) or static architecture where previous task-related parts are masked during new task training Fernando et al. (2017).

Notably, transfer learning aims to improve a (or many) target task by leveraging knowledge from a (or many) related but different source task, typically without requiring retraining from scratch (Weiss et al., 2016). In contrast, continual learning considers a sequence of tasks and aims to adapt to new tasks without forgetting previous ones. While transfer learning emphasizes performance on the target task, such as a jumpstart or reduced time in learning (Taylor & Stone, 2009), continual learning focuses on knowledge transfer, forgetting mitigation, and overall performance across all tasks.

### 2.2    Continual Reinforcement Learning

In the context of RL, continual learning involves a sequential decision-making problem over a stream of tasks where each task can be considered a stationary MDP. The most common non-stationarity in the environment is in the transition dynamics and/or reward function. Quick adaptation and building on relevant previously learned behaviors are central to the study of continual lifelong RL Khetarpal et al. (2022).

---

[1] https://github.com/XueYang0130/DGCRL.git

Recent studies have proposed various approaches to address this problem. Nagabandi et al. propose the meta-learning for online learning (MOLe) approach, which employs expectation maximization (EM) with a Chinese restaurant process (CRP) to maintain a mixture of neural dynamics models, which are meta-trained via model-agnostic meta-learning (MAML) and updated online for rapid adaptation in model-based RL (Nagabandi et al., 2018). Wang et al. introduce lifelong incremental reinforcement learning (LLIRL) using similar techniques EM with CRP to maintain a mixture model incrementally for different tasks Wang et al. (2021). Zhang et al. present the dynamics-adaptive CRL (DaCoRL) method, which applies CRP for context clustering and employs an expandable multi-head neural network to optimize the context-conditioned policy, with a knowledge distillation regularization term incorporated to mitigate catastrophic forgetting Zhang et al. (2023). However, these methods expand and fine-tune their model incrementally, incurring increasing computational and memory overhead as the number of tasks grows. Wolczyk et al. propose the ClonEx-SAC method, which employs behavioral cloning for the actor within the SAC algorithm to mitigate forgetting and enhance transfer, while reusing previous policies for faster exploration Wolczyk et al. (2022). Unfortunately, in these methods, knowledge of past tasks is typically used to shape internal optimization and rarely directly affects the agent's behavior. This may be inefficient and inflexible for knowledge reuse. For a comprehensive overview of CRL formulations and methodologies, we refer readers to the recent survey by Khetarpal et al. Khetarpal et al. (2022).

While these methods represent the state-of-the-art in CRL, they typically suffer from the knowledge decoupling problem and environment modeling problem, as discussed earlier. In contrast, our demonstration-based approach externalizes prior knowledge into a self-evolving demonstration repository that directly guides behavior and accelerates adaptation without relying on environment modeling, enabling more sufficient and efficient reuse of knowledge.

### 2.3 Demonstrations in Continual Reinforcement Learning

Many replay-based methods have been proposed to integrate prior data into memory for CRL, enabling long-term knowledge storing and balancing plasticity through on-policy learning with stability via off-policy learning (Isele & Cosgun (2018); Rolnick et al. (2019)). However, these methods often suffer from limited storage requirements and sampling bias. While using generated pseudo-data for replay (Caselles-Dupré et al. (2018), Li et al. (2021)) can address these issues, the continual training of the deep generative model or any other data generator introduces additional complexity. Furthermore, their replayed experiences primarily influence learning through optimization with limited direct guidance on behavior. In contrast, demonstrations can offer structured, trajectory-level guidance that agents can imitate directly, enabling more efficient exploration and faster adaptation.

Learning from Demonstration (LfD), also known as imitation learning, has been extensively studied in the context of autonomous robotics. Robotic agents acquire new skills by learning to imitate an expert which can be viewed as a supervised learning problem Ravichandar et al. (2019). However, its integration into CRL remains limited.

Demonstrations can be collected from trajectories, pre-trained policies, or human experts, even if sub-optimal. The key is how to effectively utilize such demonstrations to facilitate continual learning. Uchendu et al. introduced jump-start reinforcement learning (JSRL), employing sub-optimal guide policies obtained from demonstrations to gradually roll in prior policies to provide a curriculum of "good" starting states for RL exploration in single tasks Uchendu et al. (2023). The effect of the guide policy gradually diminishes with the improvement of the exploration policy. It has been successfully extended in multi-objective RL Lu et al. (2024), but its application to continual learning remains underexplored. Our work adapts JSRL from single-task to multi-task by dynamically selecting the most relevant demonstration for each task. These demonstrations, composed of trajectory-level action sequences, directly influence the agent's behavior during learning.

## 3 Problem Formulation

We formalize the MDP as $M := (\mathcal{S}, \mathcal{A}, \mathcal{T}, \gamma, R)$, where $\mathcal{S}$ and $\mathcal{A}$ represent the state and action spaces, $\mathcal{T} : S \times A \times S \to [0, 1]$ is a probabilistic transition function, $\gamma \in [0, 1)$ is a discount factor, and $R : S \times A \times S \to \mathbb{R}$ specifies the reward function. At each time step $t$, the agent observes the current state $s_t \in \mathcal{S}$, selects an action $a_t \in \mathcal{A}$ according to a policy $\pi(a_t \mid s_t)$, and transitions to the next state $s_{t+1} \sim \mathcal{T}(\cdot \mid s_t, a_t)$, receiving a scalar reward $r_{t+1} = R(s_t, a_t, s_{t+1})$. The objective of the agent is to learn a policy $\pi$ that maximizes the expected discounted cumulative reward:

$$J(\pi) = \mathbb{E}_\pi \left[ \sum_{t=0}^{\infty} \gamma^t r_t \right]. \tag{1}$$

A CRL problem can be considered as an agent interacting with a sequence of stationary tasks, i.e., $\mathcal{D} = \{\mathcal{M}_1, \mathcal{M}_2, \ldots, \mathcal{M}_N\}$, where each task $\mathcal{M}_i$, $i \in [0, N]$ symbolizes the $i$-th task the agent encounters during learning. It shares a common state–action space with all other tasks but may have distinct reward and/or transition dynamics. Each task $\mathcal{M}_i$ is assumed to last for a sufficient number of time steps to support effective agent interaction and learning.

In the long term, the agent aims to learn a policy that maximises the average return on all tasks as shown in Equation 2.

$$\mathcal{J}_{\text{CRL}}(\pi) = \frac{1}{N} \sum_{i=1}^{N} \mathbb{E}_\pi \left[ \sum_{t=0}^{\infty} \gamma^t r_t^{(i)} \right]. \tag{2}$$

Another two challenges in CRL are retaining performance on previously seen tasks to mitigate forgetting and increasing forward transfer capability using prior knowledge on new tasks. Furthermore, computational and storage constraints should be considered, indicating that it is impractical to store and replay all past experiences or retrain from scratch for each new task.

## 4 Demonstration-Guided Continual Reinforcement Learning

In this section, we provide a detailed description of DGCRL, including storing prior knowledge as demonstrations and dynamic curriculum-based exploration, with a detailed algorithm provided. Figure 1 provides a structured overview of DGCRL.

**Storing prior knowledge as external demonstrations** Unlike replay methods that typically store prior knowledge in the form of $(s, a, r, s')$ transitions, which are often mixed with online transitions for jointly training, DGCRL represent prior knowledge as external demonstrations. The demonstrations consist of action sequences, i.e., $(a_1, a_2, a_3...)$ , extracted from previously learned, potentially sub-optimal policies and are stored externally in a dedicated demonstration repository.

We begin by initializing a demonstration set $\Pi_g$ with demonstrations collected from policies trained in default environments, such as random or manually defined environments, which may differ from the environments during evaluation. This provides initial guidance for the first task $\mathcal{M}_1$ encountered by the agent.

For each task $\mathcal{M}_i$, the agent selects the most relevant demonstration $\pi_{g,i}$ from the repository, which performs the "best" (i.e., with the highest return) in the current task. $\pi_{g,i}$ serves as the prior or guide policy to initialize the agent's exploration with a relatively "good" position, enabling direct exploration guidance and fast learning within the curriculum. The return corresponding to the selected demonstration or prior policy is set as the threshold return $r_{thr,i}$.

A self-evolving strategy Lu et al. (2024) is employed that allows the demonstration repository to evolve automatically. During training, if the agent discovers a new demonstration that achieves a return higher than the current threshold $r_{thr,i}$, it is added to the demonstration repository. This supports continual learning with progressively enriched prior knowledge. In practice, the threshold return $r_{thr,i}$ can be scaled

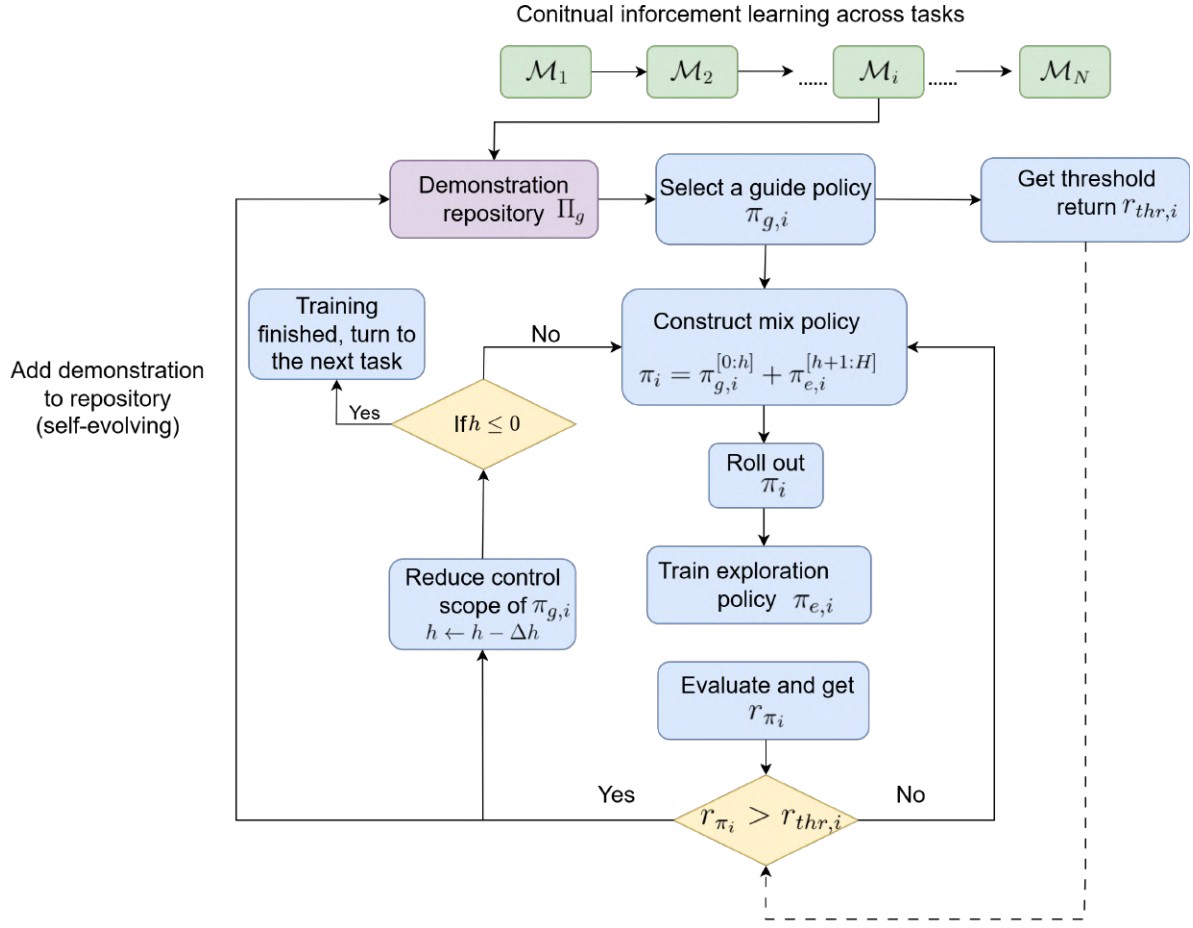

Figure 1: In DGCRL, the agent dynamically selects the most relevant demonstration from the self-evolving repository for each new task, and then follows a curriculum-based strategy to guide exploration and facilitate faster learning.

by a discount factor $\beta$, demoted as

$$\tilde{r}_{\text{thr},i}(t) = \beta_t \, r_{\text{thr},i}, \qquad \beta_t \to 1 \text{ as training progresses.}$$

With $\beta_t \in (0, 1]$ when returns are positive, and $\beta_t > 1$ (e.g., $\approx 1.3$) when returns are negative, this softens the early requirement of surpassing the given demonstration and encourages broader exploration.

**Dynamic curriculum-based exploration** After selecting the most suitable demonstration for each new task $\mathcal{M}_i$, DGCRL adopts a dynamic curriculum-based exploration strategy, extending the JSRL framework Uchendu et al. (2023) from single-task to continual learning settings.

The policy $\pi_i$ of the agent comprises two components: the guide policy $\pi_{g,i}$ and the exploration policy $\pi_{e,i}$. During training, the agent initially adheres to $\pi_{g,i}$ by imitating the actions from the demonstration for the first $h$ timesteps. Subsequently, it engages in the exploration policy $\pi_{e,i}$ for self-exploration during the remaining $H - h$ steps ($H$ denotes the task horizon, i.e., episode length). This staged approach facilitates a good starting position for RL exploration, enhancing both the initiation and simplification of the exploration process.

After the rollout of $\pi_i$, the collected data is utilized to train $\pi_{e,i}$. The agent then undertakes a single episode to evaluate the updated policy $\pi_i$ and its return $r_{\pi_i}$ is compared against the threshold return $r_{thr,i}$. If $r_{\pi_i}$ exceeds $r_{thr,i}$, the corresponding demonstration is integrated into the repository, forming a self-evolving

mechanism. At the end of each iteration, the guide horizon $h$ is reduced by $\Delta h$, progressively diminishing the influence of the guide policy as the exploration policy enhances. This iterative process continues until the end condition is met and the curriculum-based learning is repeated for each environment.

Formally, the mixed policy $\pi_i$ described above induces a task-specific return that corresponds directly to the single-task objective in Equation 1. The exploration policy $\pi_{e,i}$ is trained to maximize this return. As DGCRL applies this procedure sequentially to all tasks $\mathcal{M}_1, \mathcal{M}_2, \ldots, \mathcal{M}_N$, the method effectively optimizes the CRL objective $J_{\mathrm{CRL}}$ defined in Equation 2.

Overall, DGCRL balances stability and plasticity through three key strategies: (1) storing prior knowledge in an external demonstration repository, (2) employing a dynamic curriculum-based strategy to directly guide RL exploration and adaptation using selected demonstrations, and (3) retaining high-performing demonstrations through a self-evolving mechanism. More details are provided in Algorithm 1. Specifically, Lines 3–5 correspond to the demonstration selection phase. Lines 6–14 implement demonstration-guided exploration via curriculum learning, within which Lines 10–12 realize the self-evolving demonstration strategy.

---

**Algorithm 1** Demonstration-Guided Continual Reinforcement Learning

1: **Input:** Initial demonstration set $\Pi_g$,
2: Dynamic environment $\mathcal{D} = \{\mathcal{M}_1, \mathcal{M}_2, ..., \mathcal{M}_N\}$.
3: **for** $\mathcal{M}_i$ in $\mathcal{D}$ **do**
4:     Select the demonstration $\pi_{g,i}$ that from $\Pi_g$ that performs the best on task $\mathcal{M}_i$
5:     Set the return of $\pi_{g,i}$ as the threshold $r_{thr,i}$
6:     Initialize guide horizon $h = H$ (where $H$ is the task horizon, i.e., episode length)
7:     **while** $h \geq 0$ **do**
8:         Set mix policy $\pi_i = \pi_{g,i}^{[0:h]} + \pi_{e,i}^{[h+1:H]}$
9:         Roll out $\pi_i$, gather the experience
10:         Train exploration policy $\pi_{e,i}$
11:         Evaluate $r_{\pi_i}$ from a single rollout of $\pi_i$
12:         **if** $r_{\pi_i} > r_{thr,i}$ **then**
13:             Add the resultant demonstration of $\pi_i$ to $\Pi_g$
14:             $h \leftarrow h - \Delta h$, $\Delta h$ is the rollback span
15:         **end if**
16:     **end while**
17: **end for**

---

# 5 Experiment Settings

We evaluate DGCRL on robotics control tasks including 2D navigation and MuJoCo locomotion and compare it against various baseline methods. The environments include 50 sequential tasks with non-stationarity in reward and/or transition functions across tasks. This follows prior CRL studies Wang et al. (2021); Zhang et al. (2023), enabling fair and direct comparison. See A for DGCRL's implementation and hyperparameter details.

## 5.1 Benchmark Environments

### 5.1.1 2D Navigation

The 2D navigation environment Wang et al. (2021; 2019) refers to an agent navigating from a start position to a goal position. The state is the agent's 2D position, and the action is a 2D velocity bounded in $[-0.1, 0.1]$. The reward is the negative squared distance to the goal minus a control cost proportional to the magnitude of the action. Each episode ends upon reaching the goal (within 0.01 units) or after 100 steps. To simulate non-stationarity across tasks, three environment variants are constructed:

- **Type I**: The goal position changes, modifying the reward function.

- **Type II**: The puddle position changes, modifying the transition function. The agent should navigate to the goal while avoiding three circular puddles with different sizes. Once hitting the puddles, the agent will bounce to the previous position.

- **Type III**: Both the goal and puddle positions vary, affecting both reward and transition functions.

These three 2D navigation tasks are designed to evaluate different forms of non-stationarity in CRL. Type I only modifies the reward function while keeping the transition dynamics fixed. This setting isolates reward-level non-stationarity and tests whether DGCRL can efficiently use demonstrations to accelerate exploration in new tasks. Type II changes only the puddle locations. It alters the transition dynamics, but keeps the reward fixed. This setting tests the agent's ability to adjust its exploration strategy when the environment dynamics change. Type III changes both the goal and puddle positions, simultaneously modifying the reward and transition functions. This represents the most challenging non-stationary scenario in the 2D navigation tasks.

Figure 2 illustrates the settings, where the green circle is the start, the red circle is the goal, and the blue circle denotes puddles of varying sizes such as with radius of 0.1, 0.15, and 0.2, respectively.

More details of the 2D navigation setup can be found in prior work Wang et al. (2021; 2019). Notably, these studies overlook cases where the start or goal positions are initialized within puddles, which may render the episode ineffective, as the agent can neither move nor reach the goal. To address this, we explicitly prevent the start and goal positions from being placed inside puddles.

### 5.1.2 MuJoCo Locomotion

We further evaluate DGCRL on three standard MuJoCo locomotion benchmarks: Hopper, HalfCheetah, and Ant. Compared to the 2D navigation tasks, these environments have significantly higher dimensions of the state and action spaces and more complex dynamics. Each task involves controlling an agent to run at a specified velocity along the positive $x$-direction. The reward function consists of an "alive bonus" and a penalty component negatively correlated with the absolute deviation of the agent's current velocity $v_x$ from the target velocity $v_g$. To introduce task variations, the target velocity is randomly sampled from predefined ranges: $[0.0, 1.0]$ for Hopper, $[0.0, 2.0]$ for HalfCheetah, and $[0.0, 0.5]$ for Ant, which directly affect the reward function. Each episode starts from a fixed initial state and terminates either when the agent falls or the maximum episode length of 100 steps is reached, consistent with setups in prior work Wang et al. (2021); Zhang et al. (2023); Wang et al. (2019). Visualizations of the environments are provided in Figure 3.

### 5.2 Baselines

We employ five baseline methods, including Naive, Robust Policy, Adaptive, MAML, and LLIRL for comparative evaluation. The details of each method are as follows:

1. *Naive*: It refers to simply training a single policy sequentially over a series of tasks without explicitly accounting for task variation.

2. *Robust Policy*: It leverages domain randomization Tobin et al. (2017); Sheckells et al. (2019); Peng et al. (2018) to train a robust policy that is supposed to work across all environments, and the agent cannot infer the specific environment context.

3. *Adaptive*: It leverages a long short-term memory (LSTM) network to process a sequence of past observations and uses domain randomization to improve generalization across varying environments.

4. *MAML* Finn et al. (2017); Al-Shedivat et al. (2017): It learns a meta-policy that can quickly adapt to new tasks using only a few training samples. By exploiting the shared structure across tasks, it enables efficient fine-tuning in previously unseen environments.

5. *LLIRL* Wang et al. (2021): It uses expectation maximization (EM) with a Chinese Restaurant Process (CRP) prior to maintain and update a mixture of models for different tasks in an online manner.

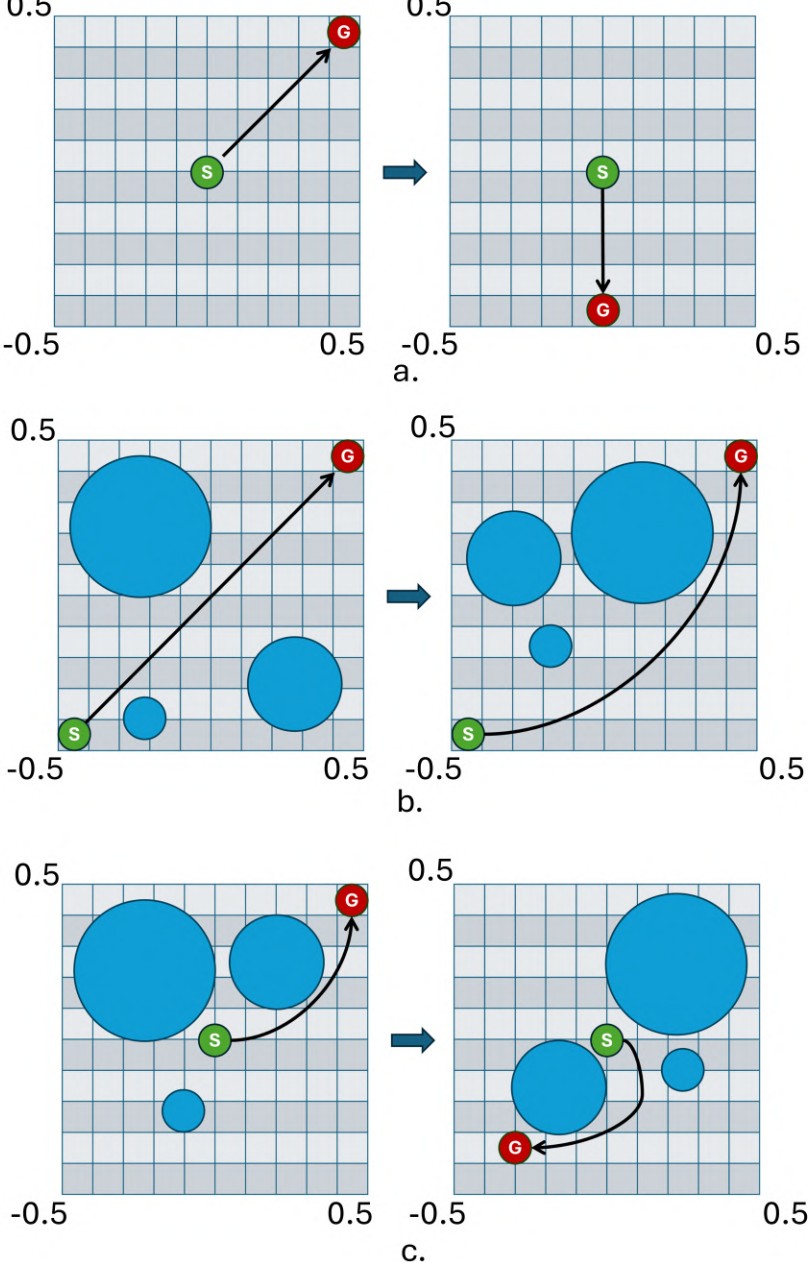

Figure 2: 2D Navigation (a) Type 1 (v1): Only the goal position changes. (b) Type 2 (v2): Only the puddle positions change. (c) Type 3 (v3): Both the puddle and goal positions change.

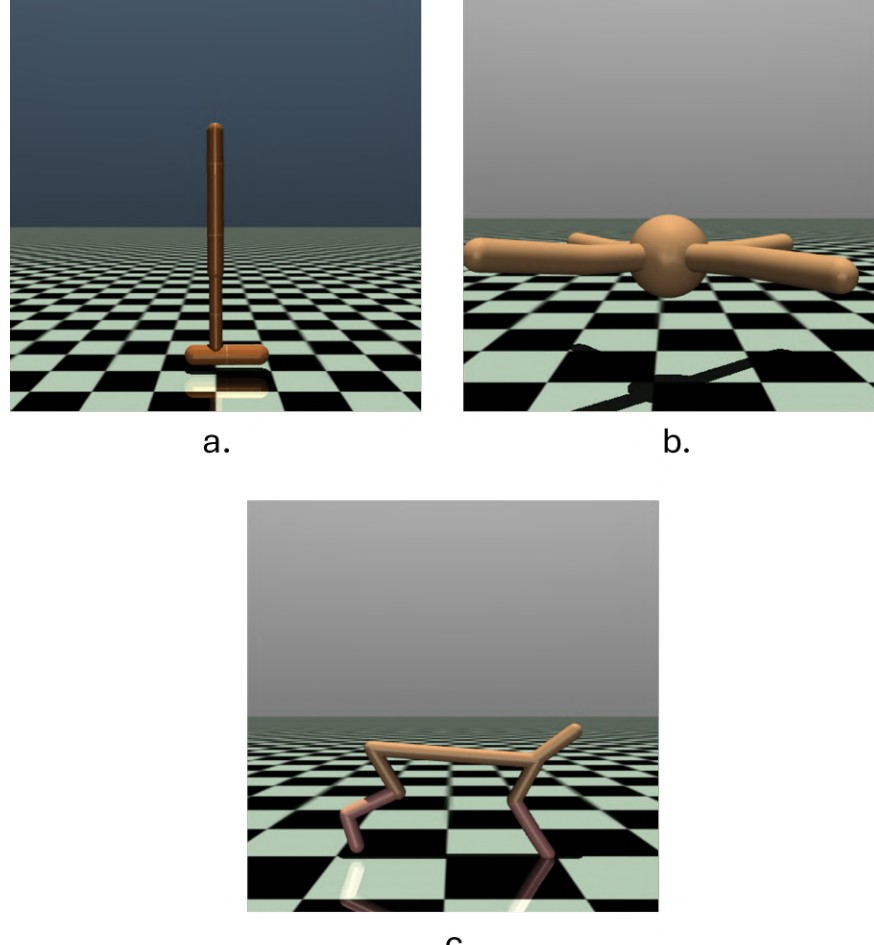

Figure 3: Mujoco locomotion (a) Hopper, with state dimension $|\mathcal{S}| = 11$, action dimension $|\mathcal{A}| = 3$, and reward function $r = 1 - 4|v_x - v_g|$ (b) Ant, $|\mathcal{S}| = 111$, $|\mathcal{A}| = 8$, and $r = 1 - 3|v_x - v_g|$ (c) Halfcheetah, $|\mathcal{S}| = 20$, $|\mathcal{A}| = 6$, and $r = -|v_x - v_g|$.

### 5.3 Metrics

Four metrics are utilized to evaluate the effectiveness of DGCRL. The first three metrics are adapted from those used in Wolczyk et al. (2022), which measure the average performance, forward transfer capability, and forgetting. The fourth metric, adapted from Wang et al. (2021), measures the average return over episodes for different tasks. The performance (i.e. return) for task $i$ at time $t$ is termed as $p_i(t)$, and each of the $N$ tasks is trained for $\Delta$ steps. Therefore, the total number of steps for all tasks is $T = N \cdot \Delta$.

**Average Performance** (AP): The average performance at time $t$ is defined as:

$$P(t) := \frac{1}{N} \sum_{i=1}^{N} p_i(t). \tag{3}$$

We evaluate the final average performance, denoted as $P(N)$.

**Forward Transfer** (FT): FT is used to measure the agent's ability to leverage knowledge acquired from previous tasks to improve the learning process of new tasks. The FT for task $i$ is defined as the normalized area of the gap between the candidate method's training curve ($AUC_i$) and the reference training curve ($AUC_i^b$). Denoting the reference performance as $p_i^b$, the forward transfer on task $i$, $FT_i$, is defined as:

$$FT_i := \frac{AUC_i - AUC_i^b}{max[A, AUC^b] - AUC_i^b}, \tag{4}$$

where the candidate method's training curve is:

$$AUC_i := \frac{1}{\Delta} \int_{(i-1)\cdot\Delta}^{i\cdot\Delta} (p_i(t) + C)dt, \tag{5}$$

the reference training curve is:

$$AUC_i^b := \frac{1}{\Delta} \int_{0}^{\Delta} (p_i^b(t) + C)dt, \tag{6}$$

and $A := \{AUC_i \mid i = 1, 2, \ldots, N\}$. For all tasks, the average forward transfer is defined as:

$$FT = \frac{1}{N} \sum_{i=1}^{N} FT_i. \tag{7}$$

Without compromising the fairness of comparisons, we adapt the metric used in the original works and use return values rather than a success rate as the performance metric. The value of $p_i(t)$ is consequently not necessarily restricted to the range between 0 and 1. We utilize the maximum area among the training curves to appropriately normalize the FT metric rather than value 1 as in the original works. Additionally, given that return values can be negative in some environments, such as the half-cheetah, we introduce a constant factor $C$, to ensure that the calculated area remains non-negative. This adjustment is necessary to maintain the mathematical integrity and relevance of our performance assessment.

**Forgetting**: Forgetting is the decline in a method's performance on previously learned tasks after learning new tasks. Take task $i$ as an example, the forgetting is measured as the performance drop between the moment when the agent just finishes the training on task $i$ and the moment when the whole training process is over, denoted as:

$$F_i = p_i(i \cdot \Delta) - p_i(T). \tag{8}$$

The forgetting metric over the whole training process is therefore:

$$F = \frac{1}{N} \sum_{i=1}^{N} F_i. \tag{9}$$

**Average return over episodes for different tasks** (Avg. Return): This metric is to assess the policy's learning performance across various tasks. The average return for the total $N$ tasks in the $j$-th episode is defined as:

$$r_j = \frac{1}{N} \sum_{i=1}^{N} r_{i,j}. \tag{10}$$

This metric is represented in the form of a learning curve. Compared with the metric AP, which evaluates the final average performance across all tasks, the metric Avg. Return evaluates the policy's learning dynamics by assessing the average return per episode, focusing on the training process.

## 6 Experimental Results

We evaluate DGCRL on 2D Navigation and MuJoCo locomotion environments, comparing it to baseline methods using the metrics described above. We further conduct a sensitivity analysis to assess the impact of demonstration quantity on performance. Two ablation studies are also included: one examining the effect of resetting different components of the TD3 actor-critic architecture, and the other isolating the benefit of combining self-evolving demonstration guidance with curriculum-based exploration. All results are averaged over multiple random seeds. We use stratified bootstrap confidence intervals and interquartile mean (IQM), following the recommendation in Agarwal et al. (2021) for robust and reliable few-run evaluations.

### 6.1 Main Experiment

The experimental results of *Average Performance*, *Forward Transfer*, and *Forgetting* for three navigation tasks (Navigation v1, Navigation v2, and Navigation v3) and three locomotion tasks (Hopper, Ant, and Half Cheetah) are presented in Table 1, where the arrows indicate the direction of improvement and the best performance for each metric is highlighted in bold.

DGCRL demonstrates consistently superior performance over baselines across almost all environments and metrics, achieving the best average performance, the highest forward transfer, and the lowest forgetting. For instance, in the Hopper environment, DGCRL achieves the highest average performance, significantly outperforming the second-best MAML and the third-best Adaptive. It also attains the highest forward transfer, greatly surpassing the second-best Robust Policy and the third-best MAML. Furthermore, DGCRL achieves the lowest forgetting, substantially reducing forgetting compared to the second-best Adaptive and the third-best MAML. This indicates the significant advantages of DGCRL in balancing stability and plasticity in dynamic environments.

Notably, DGCRL exhibits negative forgetting. In conventional continual learning, the forgetting metric implicitly assumes that each task achieves its best performance at the end of training. However, this does not always hold in CRL under limited training budgets, where convergence often leads to suboptimal policies. Due to the stochasticity of exploration and the instability of function approximation, better solutions may temporarily appear during training but fail to be retained, which is termed as "derailment" in the RL literature Ecoffet et al. (2021). This implies that superior demonstrations may arise during the learning process. Furthermore, when tasks share similar dynamics, exploration in later tasks may also uncover solutions that achieve higher returns on earlier tasks. In our method, through the self-evolving demonstration strategy, such high-quality demonstrations are continuously stored in an external repository, and during evaluation the agent can retrieve the most suitable demonstration to directly guide its behavior. Consequently, negative forgetting values may occur. Overall, these findings highlight both the limitations of the conventional forgetting metric in CRL and the importance of storing prior knowledge continuously that can be easily retrieved.

The only exception is the forward transfer metric on the Ant environment, where DGCRL slightly lags behind MAML but remains competitive. This can be attributed to MAML's ability to effectively capture cross-task knowledge during meta-training, allowing for rapid adaptation and relatively good performance on new tasks. However, since MAML is not explicitly designed to retain task-specific knowledge, its performance in terms of forgetting is less impressive.

Table 1: Summary of Evaluation - Main Experiment

| Baselines | Metric | | | | | |
|---|---|---|---|---|---|---|
| | AP↑ | FT↑ | Forgetting↓ | AP↑ | FT↑ | Forgetting↓ |
| | Navigation v1 | | | Navigation v2 | | |
| Naive | $-45.93^{+5.36}_{-6.31}$ | $0.43^{+0.17}_{-0.17}$ | $23.28^{+4.78}_{-3.7}$ | $-66.11^{+21.57}_{-13.46}$ | $0.51^{+0.06}_{-0.15}$ | $21.37^{+23.93}_{-11.25}$ |
| Robust Policy | $-54.66^{+1.06}_{-3.26}$ | $0.46^{+0.07}_{-0.11}$ | $27.49^{+1.7}_{-0.7}$ | $-61.07^{+2.14}_{-4.5}$ | $0.27^{+0.07}_{-0.04}$ | $19.4^{+2.93}_{-1.83}$ |
| Adaptive | $-78.64^{+0.92}_{-1.11}$ | $-0.02^{+0.19}_{-0.22}$ | $22.2^{+0.84}_{-1.08}$ | $-98.18^{+2.21}_{-1.69}$ | $-0.41^{+0.16}_{-0.08}$ | $22.24^{+3.15}_{-1.4}$ |
| MAML | $-43.18^{+0.5}_{-0.9}$ | $0.62^{+0.07}_{-0.09}$ | $22.91^{+0.46}_{-0.27}$ | $-41.71^{+0.26}_{-0.41}$ | $0.55^{+0.04}_{-0.02}$ | $10.52^{+1.07}_{-0.79}$ |
| LLIRL | $-47.08^{+2.67}_{-2.21}$ | $0.6^{+0.09}_{-0.09}$ | $31.97^{+1.84}_{-2.59}$ | $-50.27^{+16.37}_{-28.51}$ | $0.45^{+0.13}_{-0.13}$ | $4.35^{+20.43}_{-10.49}$ |
| DGCRL (ours) | $\mathbf{-6.74^{+0.1}_{-0.13}}$ | $\mathbf{0.82^{+0.03}_{-0.05}}$ | $\mathbf{-1.31^{+0.32}_{-0.94}}$ | $\mathbf{-7.72^{+0.02}_{-0.03}}$ | $\mathbf{0.69^{+0.04}_{-0.05}}$ | $\mathbf{-15.52^{+6.26}_{-14.24}}$ |
| | Navigation v3 | | | Hopper | | |
| Naive | $-47.83^{+1.69}_{-1.99}$ | $-0.03^{+0.13}_{-0.18}$ | $22.72^{+3.2}_{-1.84}$ | $-12.28^{+7.98}_{-16.28}$ | $-0.14^{+0.1}_{-0.01}$ | $35.65^{+31.85}_{-22.72}$ |
| Robust Policy | $-57.39^{+1.84}_{-2.57}$ | $-0.15^{+0.2}_{-0.34}$ | $27.65^{+1.63}_{-0.82}$ | $-4.35^{+0.75}_{-4.04}$ | $-0.06^{+0.19}_{-0.29}$ | $40.51^{+16.55}_{-18.16}$ |
| Adaptive | $-78.89^{+1.64}_{-0.46}$ | $-1.06^{+0.38}_{-0.46}$ | $20.77^{+0.6}_{-0.96}$ | $-3.97^{+0.76}_{-1.63}$ | $-0.12^{+0.05}_{-0.11}$ | $34.22^{+13.09}_{-10.99}$ |
| MAML | $-45.7^{+0.24}_{-0.42}$ | $0.14^{+0.18}_{-0.18}$ | $23.05^{+0.53}_{-0.19}$ | $-3.12^{+0.26}_{-6.23}$ | $-0.08^{+0.02}_{-0.21}$ | $35.72^{+5.25}_{-11.14}$ |
| LLIRL | $-31.61^{+0.56}_{-0.19}$ | $0.41^{+0.1}_{-0.17}$ | $14.63^{+0.8}_{-0.56}$ | $-25.84^{+10.88}_{-19.74}$ | $-0.14^{+0.1}_{-0.01}$ | $60.66^{+27.08}_{-16.51}$ |
| DGCRL (ours) | $\mathbf{-3.25^{+0.28}_{-0.48}}$ | $\mathbf{0.65^{+0.04}_{-0.06}}$ | $\mathbf{-9.46^{+6.26}_{-12.2}}$ | $\mathbf{93.85^{+0.14}_{-0.12}}$ | $\mathbf{0.8^{+0.02}_{-0.03}}$ | $\mathbf{-3.47^{+1.08}_{-3.17}}$ |
| | Ant | | | Half Cheetah | | |
| Naive | $38.32^{+3.59}_{-1.71}$ | $-1.64^{+1.16}_{-3.18}$ | $-9.77^{+25.91}_{-28.3}$ | $-78.86^{+2.8}_{-5.47}$ | $-4.13^{+0.37}_{-0.21}$ | $53.0^{+3.71}_{-2.23}$ |
| Robust Policy | $-12.51^{+30.03}_{-5.7}$ | $-2.48^{+1.82}_{-2.76}$ | $19.33^{+37.15}_{-17.74}$ | $-81.63^{+8.46}_{-9.47}$ | $-4.49^{+0.61}_{-2.34}$ | $49.83^{+2.17}_{-1.97}$ |
| Adaptive | $32.05^{+8.0}_{-34.1}$ | $-0.8^{+0.87}_{-1.01}$ | $32.85^{+8.61}_{-7.69}$ | $-76.98^{+2.37}_{-5.15}$ | $-3.96^{+0.6}_{-0.5}$ | $46.27^{+1.48}_{-3.02}$ |
| MAML | $45.83^{+3.58}_{-6.64}$ | $\mathbf{-0.34^{+0.48}_{-0.33}}$ | $21.77^{+3.62}_{-5.08}$ | $-66.46^{+1.11}_{-3.11}$ | $-0.34^{+0.48}_{-0.33}$ | $42.56^{+10.33}_{-11.55}$ |
| LLIRL | $6.52^{+45.1}_{-31.3}$ | $-2.99^{+1.84}_{-2.28}$ | $3.2^{+5.52}_{-29.28}$ | $-27.07^{+1.03}_{-0.68}$ | $-3.25^{+1.34}_{-0.76}$ | $8.39^{+4.04}_{-1.19}$ |
| DGCRL (ours) | $\mathbf{80.25^{+4.83}_{-8.35}}$ | $-0.5^{+0.49}_{-0.41}$ | $\mathbf{-12.97^{+3.88}_{-6.71}}$ | $\mathbf{-3.58^{+0.17}_{-0.63}}$ | $\mathbf{0.25^{+0.04}_{-0.02}}$ | $\mathbf{-3.68^{+0.68}_{-0.99}}$ |

LLIRL performs reasonably well in simple navigation tasks but exhibits a significant decline in more complex tasks, such as Hopper and Ant, raising concerns about its robustness, a critical requirement in safe RL. Unfortunately, we were unable to fully replicate the original work's reported results.

Interestingly, we observe that despite lacking specific mechanisms for environment identification or domain randomization, the *Naive* method remains competitive with the enhanced baselines *Robust Policy* and *Adaptive*, and even outperforms them in certain metrics across some environments. It indicates that domain randomization and using LSTM to retain the knowledge of past tasks are not silver-bullet solutions for handling CRL tasks.

Figure 4 illustrates the results of *Average return over episodes* for different tasks. DGCRL consistently outperforms baselines, except in the Ant environment where it performs comparably to MAML. We note that a brief initial drop is observed in some tasks (e.g., Ant and Navigation), which can be attributed to the transition from demonstration-guided exploration to self-exploration. At the beginning, the agent fully imitates the demonstration to achieve rapid early gains; performance then temporarily declines as reliance on demonstrations is reduced and self-exploration increases. This leads to temporary fluctuation, but as exploration policy improves, training performance correspondingly recovers and rapidly increases. These results highlight the benefit of demonstrations in providing a strong initial policy and facilitating faster learning. Furthermore, we observe fluctuations in policy performance during the later stages of training in environments like Navigation v2 and v3. This instability may stem from weak correlations between tasks.

## 6.2   Sensitivity Analysis

This experiment evaluates the impact of different proportions of initial demonstrations, i.e., 20%, 60%, and 100% of the full set[2] (sampled randomly), on CRL performance in different benchmark environments. Note that varying the size of the initial demonstration set implicitly changes the demonstration quality: a smaller set covers fewer scenario variations and therefore represents a more sub-optimal demonstration pool. Moreover, a larger initial demonstration set increases the likelihood of selecting a higher-quality demonstration for each task. The performance differences thus also reflect DGCRL's robustness to sub-optimal initial demonstrations.

Table 2 presents the results of *Average Performance*, *Forward Transfer*, and *Forgetting*. It shows that DGCRL equipped with the full demonstration set outperforms the others in average performance and forward transfer in most environments. For instance, in Hopper, the full demonstration set leads to the highest average performance and forward transfer, significantly surpassing the reduced sets. Although it ranks second in average performance for Navigation v3 and in forward transfer for Ant, its performance remains highly competitive.

Additionally, we note that the number of demonstrations has limited impact on the forgetting metric, and some smaller demonstration sets even exhibit better results. This may be attributed to the weaker initial conditions associated with a smaller demonstration set, which results in lower early performance. As training progresses, these models undergo substantial improvement, making the reduction in forgetting performance appear more pronounced. In contrast, models with the full demonstration set start closer to their peak performance due to the comprehensive initial guidance, leading to less noticeable forgetting.

Figure 5 presents the *Average return over episodes*, where the green curve represents training with the full demonstration set, and the blue and red curves denote reduced sets. Overall, more demonstrations lead to better initialization, faster convergence, and improved final performance. An exception is observed in the Ant environment, where additional demonstrations primarily benefit early learning but have limited long-term impact. This is likely due to the environment's complexity and limited marginal gain from extra trajectories. Nevertheless, even with minimal demonstrations, DGCRL consistently outperforms all baselines.

---

[2]The full set denotes the number of demonstrations employed in the main experiments.

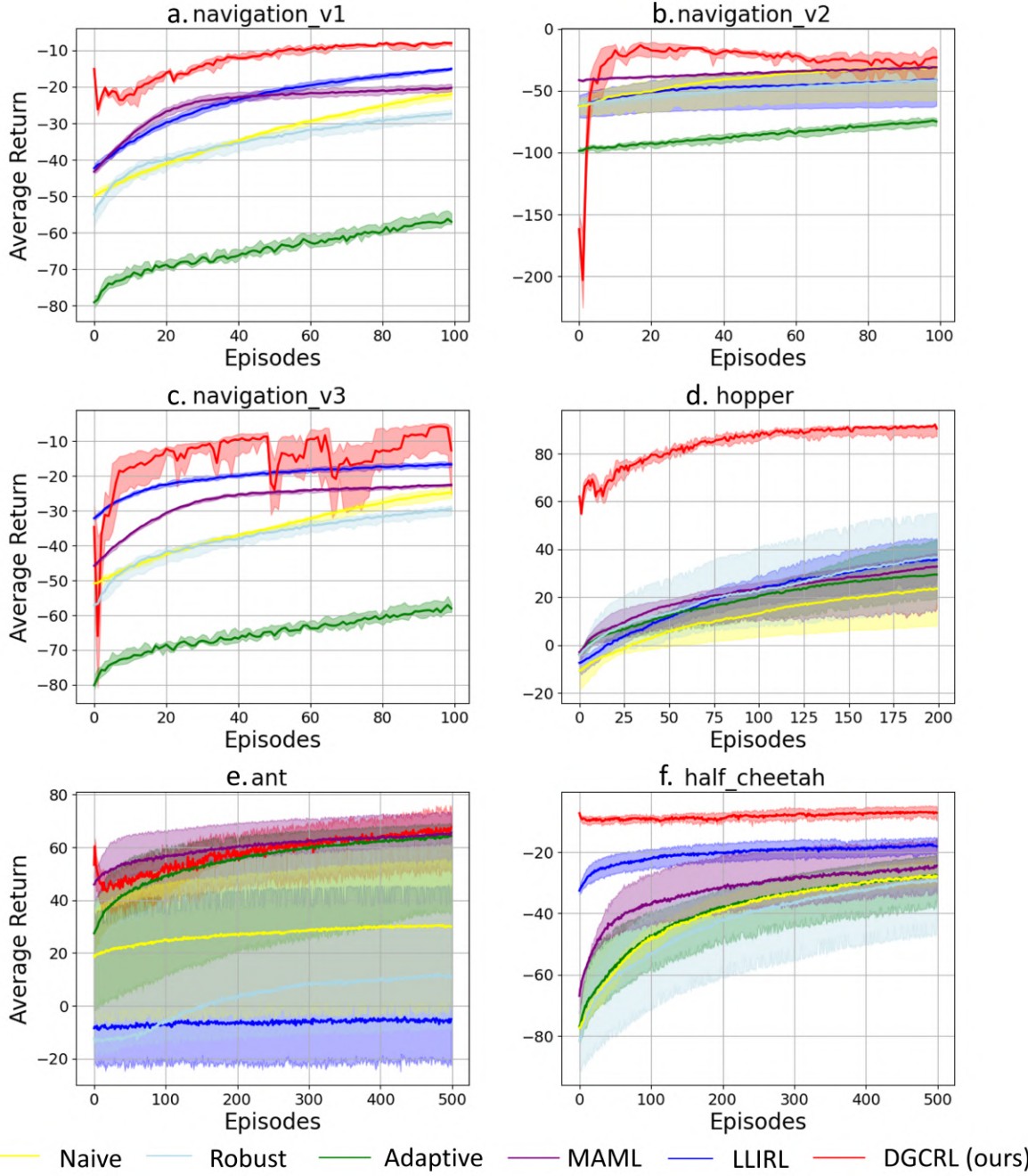

Figure 4: Average return per episode - Main Experiment

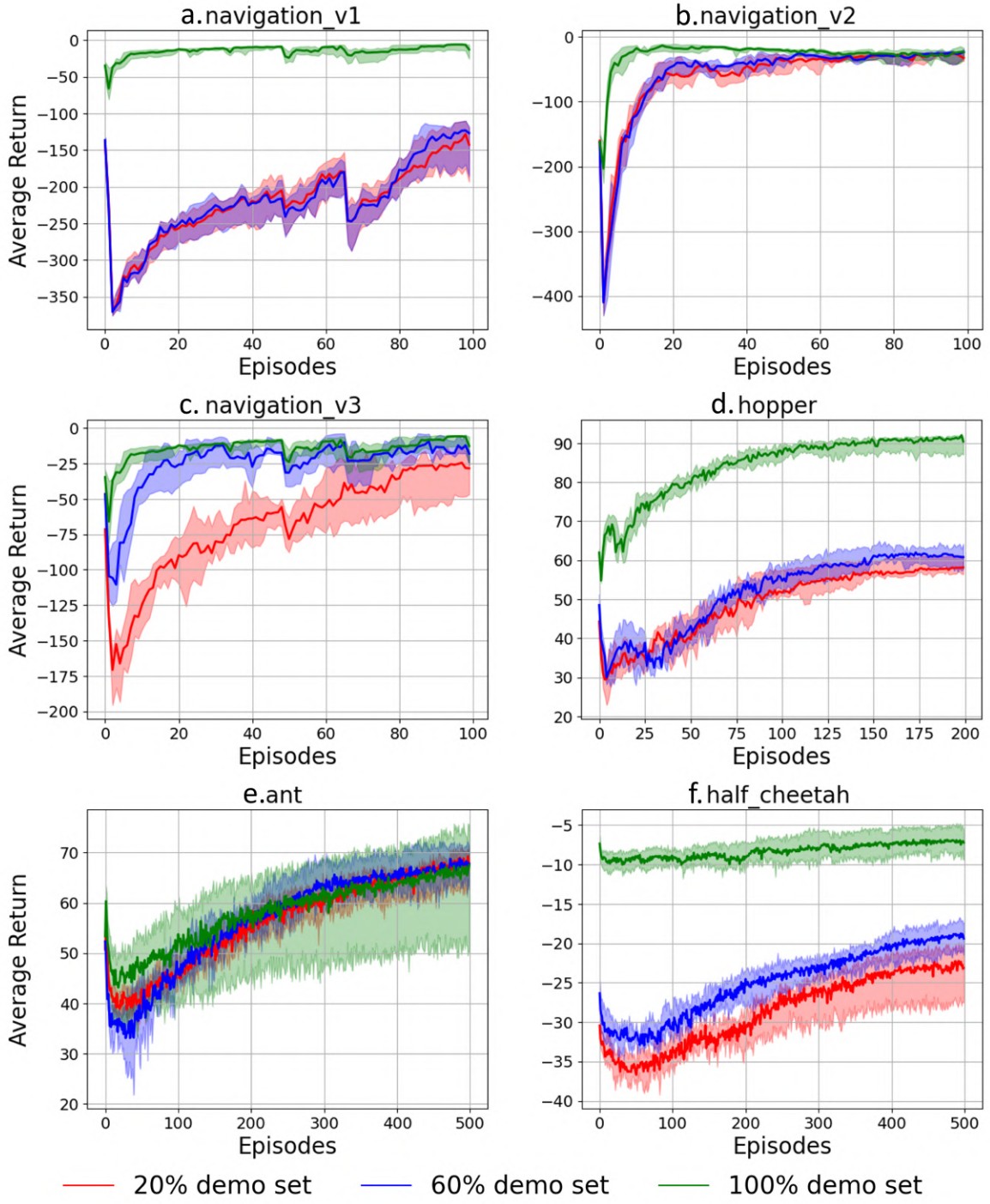

Figure 5: Average return per episode - Sensitivity Analysis

Table 2: Summary of Evaluation - Sensitivity Analysis

| Baselines | Metric | | | | | |
|---|---|---|---|---|---|---|
| | AP↑ | FT↑ | Forgetting↓ | AP↑ | FT↑ | Forgetting↓ |
| | **Navigation v1** | | | **Navigation v2** | | |
| 20% demo set | $-23.01^{+0.63}_{-1.75}$ | $-1.0^{+0.39}_{-1.29}$ | $\mathbf{-119.8^{+22.56}_{-48.89}}$ | $-11.6^{+3.83}_{-1.98}$ | $0.24^{+0.11}_{-0.12}$ | $\mathbf{-19.15^{+8.19}_{-8.09}}$ |
| 60% demo set | $-22.97^{+0.59}_{-1.79}$ | $-1.44^{+0.74}_{-1.19}$ | $-103.99^{+8.87}_{-49.95}$ | $-7.86^{+0.08}_{-0.06}$ | $0.31^{+0.03}_{-0.05}$ | $-15.23^{+4.44}_{-9.96}$ |
| 100% demo set | $\mathbf{-6.74^{+0.1}_{-0.13}}$ | $\mathbf{0.82^{+0.03}_{-0.05}}$ | $-1.31^{+0.32}_{-0.94}$ | $\mathbf{-7.72^{+0.02}_{-0.03}}$ | $\mathbf{0.69^{+0.03}_{-0.05}}$ | $-15.52^{+6.26}_{-14.24}$ |
| | **Navigation v3** | | | **Hopper** | | |
| 20% demo set | $-8.87^{+1.47}_{-0.73}$ | $-0.68^{+0.47}_{-0.37}$ | $\mathbf{-20.04^{+11.43}_{-18.06}}$ | $64.4^{+2.35}_{-1.0}$ | $0.28^{+0.05}_{-0.07}$ | $-5.94^{+1.05}_{-1.45}$ |
| 60% demo set | $\mathbf{-3.18^{+0.57}_{-0.55}}$ | $0.47^{+0.13}_{-0.83}$ | $-14.79^{+0.94}_{-9.02}$ | $67.21^{+1.77}_{-1.08}$ | $0.32^{+0.06}_{-0.09}$ | $\mathbf{-6.91^{+2.79}_{-2.17}}$ |
| 100% demo set | $-3.25^{+0.32}_{-0.41}$ | $\mathbf{0.65^{+0.04}_{-0.07}}$ | $-9.46^{+6.26}_{-12.2}$ | $\mathbf{93.85^{+0.13}_{-0.16}}$ | $\mathbf{0.8^{+0.02}_{-0.03}}$ | $-3.47^{+1.08}_{-3.21}$ |
| | **Ant** | | | **Half Cheetah** | | |
| 20% demo set | $77.15^{+1.28}_{-1.88}$ | $-0.58^{+0.62}_{-0.39}$ | $-9.41^{+1.99}_{-2.45}$ | $-14.58^{+1.13}_{-1.69}$ | $-2.44^{+0.41}_{-0.34}$ | $\mathbf{-8.71^{+2.38}_{-2.51}}$ |
| 60% demo set | $77.93^{+1.71}_{-1.48}$ | $\mathbf{-0.49^{+0.64}_{-0.48}}$ | $-10.07^{+3.2}_{-4.01}$ | $-13.1^{+1.02}_{-1.46}$ | $-1.97^{+0.45}_{-0.25}$ | $-6.3^{+1.11}_{-0.59}$ |
| 100% demo set | $\mathbf{80.25^{+4.83}_{-7.45}}$ | $-0.5^{+0.49}_{-0.41}$ | $\mathbf{-12.97^{+3.59}_{-8.67}}$ | $\mathbf{-3.58^{+0.73}_{-0.75}}$ | $\mathbf{0.25^{+0.07}_{-0.15}}$ | $-3.68^{+1.33}_{-1.66}$ |

## 6.3 Ablation Study

To better understand the contribution of the core components in DGCRL, we conduct two ablation studies: one examining the effect of different resetting strategies in the actor-critic architecture, and another isolating the role of curriculum-guided exploration by comparing DGCRL with trajectory-level replay baselines that only access past demonstrations.

### 6.3.1 Impact of Resetting Strategies

Since DGCRL builds on the TD3 actor-critic framework, we analyze the effect of different resetting strategies: actor-only, critic-only, and joint reset on performance. A comparative analysis of the *Average Performance*, *Forward Transfer*, and *Forgetting* metrics is shown in Table 3.

We observe that resetting both the actor and critic generally yields better performance, particularly in average return and forward transfer. While not always achieving the lowest forgetting, it remains negative and highly competitive, particularly when compared with baseline results in Table 1. These findings support our choice to reset both components in the main experiments within the DGCRL actor-critic framework.

The *Average return per episode*, as shown in Figure 6, highlights the training performance for different reset strategies: the green curve represents resetting both the actor and critic, while the blue and red curves correspond to resetting the critic and actor independently, respectively. Overall, resetting both components consistently yields superior training results and facilitates quicker adaptation to new challenges.

### 6.3.2 Comparison with Trajectory-level Replay Baselines

We additionally include two trajectory-level replay baselines for comparison:

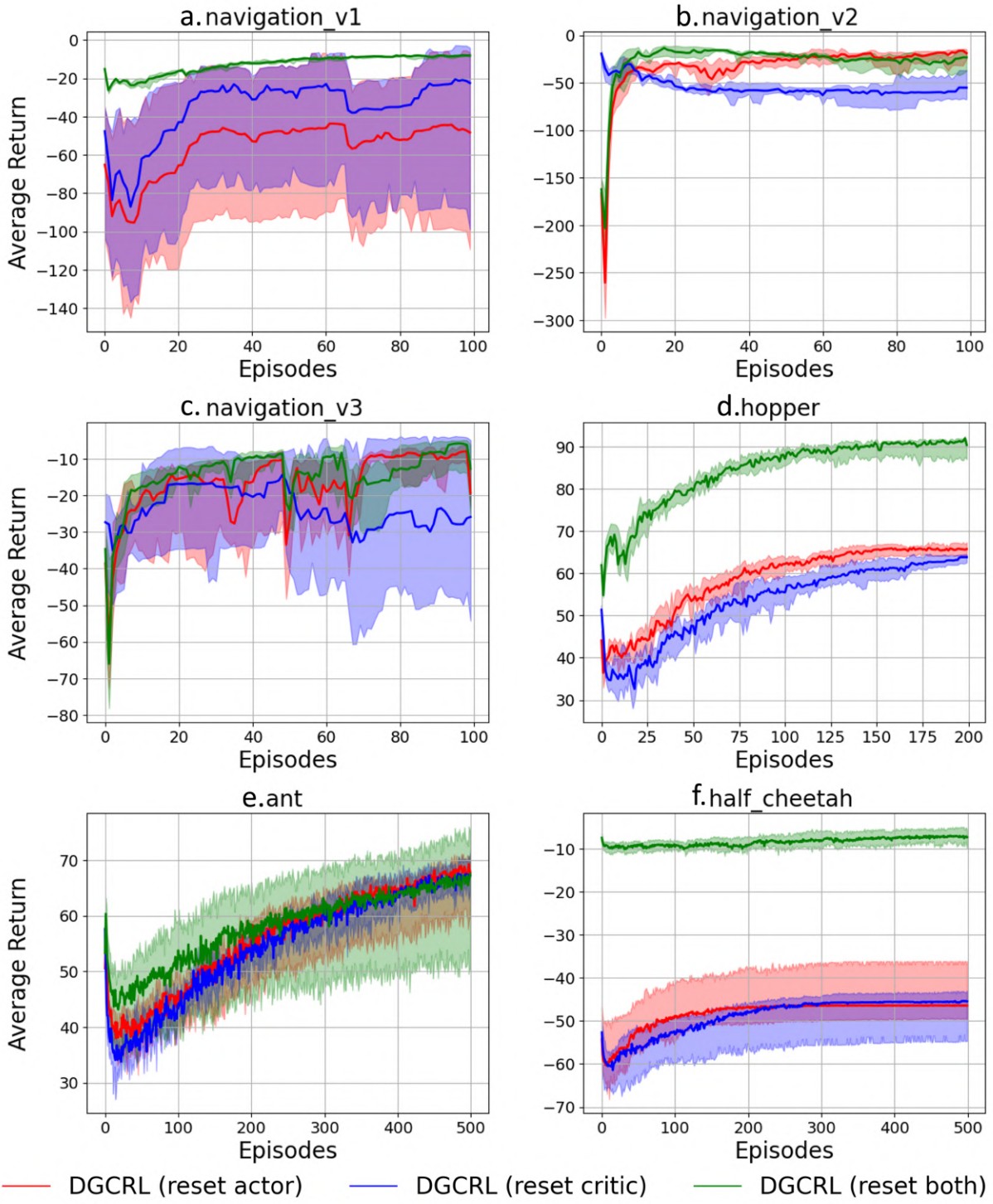

Figure 6: Average return per episode - Impact of Resetting Strategies

Table 3: Summary of Evaluation - Impact of Resetting Strategies

| Baselines | Metric | | | | | |
|---|---|---|---|---|---|---|
| | AP↑ | FT↑ | Forgetting↓ | AP↑ | FT↑ | Forgetting↓ |
| | Navigation v1 | | | Navigation v2 | | |
| Reset actor | $-21.32^{+3.75}_{-1.51}$ | $0.57^{+0.13}_{-0.29}$ | $\mathbf{-28.17^{+40.25}_{-59.18}}$ | $-7.81^{+0.06}_{-0.04}$ | $0.65^{+0.09}_{-0.06}$ | $-10.94^{+4.33}_{-5.11}$ |
| Reset critic | $-19.98^{+3.75}_{-2.89}$ | $0.59^{+0.22}_{-0.26}$ | $-3.65^{+18.98}_{-72.85}$ | $-7.76^{+0.06}_{-0.01}$ | $0.43^{+0.05}_{-0.06}$ | $\mathbf{-47.36^{+17.08}_{-12.24}}$ |
| Reset both | $\mathbf{-6.74^{+0.1}_{-0.11}}$ | $\mathbf{0.82^{+0.03}_{-0.05}}$ | $-1.31^{+0.32}_{-0.94}$ | $\mathbf{-7.72^{+0.02}_{-0.03}}$ | $\mathbf{0.69^{+0.04}_{-0.05}}$ | $-15.52^{+6.26}_{-14.24}$ |
| | Navigation v3 | | | Hopper | | |
| Reset actor | $-3.42^{+0.18}_{-0.5}$ | $0.49^{+0.13}_{-0.03}$ | $-15.75^{+7.04}_{-4.53}$ | $71.16^{+0.26}_{-1.08}$ | $0.39^{+0.06}_{-0.03}$ | $-5.59^{+1.71}_{-0.55}$ |
| Reset critic | $-3.46^{+0.79}_{-1.18}$ | $0.43^{+0.37}_{-0.92}$ | $\mathbf{-21.91^{+19.63}_{-28.64}}$ | $69.24^{+0.99}_{-0.43}$ | $0.32^{+0.03}_{-0.02}$ | $\mathbf{-5.94^{+1.37}_{-0.83}}$ |
| Reset both | $\mathbf{-3.25^{+0.32}_{-0.41}}$ | $\mathbf{0.65^{+0.04}_{-0.07}}$ | $-9.46^{+6.26}_{-12.2}$ | $\mathbf{93.85^{+0.13}_{-0.16}}$ | $\mathbf{0.8^{+0.02}_{-0.03}}$ | $-3.47^{+1.08}_{-3.21}$ |
| | Ant | | | Half Cheetah | | |
| Reset actor | $77.15^{+1.1}_{-2.9}$ | $-0.65^{+0.69}_{-0.39}$ | $-9.41^{+1.62}_{-4.13}$ | $-26.14^{+2.76}_{-3.21}$ | $-5.4^{+1.61}_{-0.56}$ | $\mathbf{-20.01^{+7.32}_{-0.57}}$ |
| Reset critic | $76.47^{+1.96}_{-1.98}$ | $-0.51^{+0.42}_{-0.43}$ | $-9.25^{+2.81}_{-2.31}$ | $-28.66^{+4.38}_{-7.72}$ | $-5.59^{+0.91}_{-1.15}$ | $-18.18^{+4.11}_{-2.31}$ |
| Reset both | $\mathbf{80.25^{+4.6}_{-8.35}}$ | $\mathbf{-0.5^{+0.49}_{-0.41}}$ | $\mathbf{-12.97^{+3.59}_{-8.67}}$ | $\mathbf{-3.58^{+0.73}_{-0.75}}$ | $\mathbf{0.25^{+0.07}_{-0.15}}$ | $-3.68^{+1.33}_{-1.66}$ |

1. *Initial Trajectory Replay (ITR)*: This baseline reuses only the trajectory-level demonstrations in the initial demonstration repository. For each task, it selects and executes the best demonstration from this fixed set. No new demonstrations collected during training are incorporated.

2. *Evolving Trajectory Replay (ETR)*: This baseline reuses all demonstrations collected by the DGCRL agent throughout the entire training process, where the demonstration repository keeps self-evolving, i.e., newly discovered high-return demonstrations are continuously stored into the initial set. For each task, it evaluates all available demonstrations from previous tasks and executes the best one.

Note that the replay baselines do not learn a policy. Instead, they simply select the best demonstration for each task and execute it directly, allowing us to isolate the effect of the curriculum mechanism in DGCRL. Since ITR and ETR perform no task-wise adaptation or optimization, metrics such as forward transfer and forgetting are not applicable. We therefore report only the Average Performance (AP) metric for these baselines.

Across all six benchmarks in Table 4, DGCRL achieves the highest average performance, highlighting the benefits of direct demonstration guidance together with a self-evolving demonstration repository and curriculum-based exploration. ITR performs well only when the initial demonstrations are near-optimal. Since it never incorporates new demonstrations, its performance is limited by the quality of the initial demonstrations. ETR benefits from the self-evolving demonstration repository, as the agent discovers high quality demonstrations during training. However, both ITR and ETR underperform DGCRL, showing that simply accessing past demonstrations only is insufficient for CRL. Additionally, the average performance of ITR and ETR generally exceeds that of the other baseline methods in Table 1, indicating that leveraging demonstrations to directly guide the agent's behavior provides a clear advantage.

These results confirm that DGCRL's performance advantages arise not just from access to stored demonstrations, but from its full learning pipeline: direct demonstration guidance, self-evolving demonstration repository, and curriculum-based exploration.

Table 4: Average Performance Comparison: DGCRL vs. Replay Baselines

| Baselines | Navigation v1 | Navigation v2 | Navigation v3 |
|---|---|---|---|
| ITR | $-30.20$ | $-8.07$ | $-6.57$ |
| ETR | $-20.13^{+1.19}_{-1.23}$ | $-7.95^{+0.05}_{-0.04}$ | $-6.31^{+0.37}_{-0.42}$ |
| DGCRL (ours) | $-6.74^{+0.10}_{-0.13}$ | $-7.72^{+0.02}_{-0.03}$ | $-3.25^{+0.28}_{-0.48}$ |

| Baselines | Hopper | Ant | Half Cheetah |
|---|---|---|---|
| ITR | $62.90$ | $62.47$ | $-87.60$ |
| ETR | $67.48^{+0.43}_{-0.64}$ | $72.06^{+1.61}_{-1.25}$ | $-32.71^{+2.19}_{-2.49}$ |
| DGCRL (ours) | $93.85^{+0.14}_{-0.12}$ | $80.25^{+4.83}_{-8.35}$ | $-3.58^{+0.17}_{-0.63}$ |

## 7 Conclusion

Existing continual learning methods apply prior knowledge implicitly through neural network optimization without direct guidance on the agent's behavior, which may limit knowledge reuse and learning efficiency. Moreover, some methods such as (Nagabandi et al., 2018; Wang et al., 2021; Zhang et al., 2023) require modeling different environments separately, which increases computational cost and introduces additional uncertainty. In contrast, our proposed method, i.e., demonstration-guided continual reinforcement learning (DGCRL), solves the CRL problem by leveraging selected demonstrations to directly guide the agent's exploration behavior without environment modeling.

Specifically, DGCRL utilizes external demonstrations to store prior knowledge and guide the agent's exploration via a dynamic curriculum strategy: for each new task, it selects the most relevant demonstration and uses it as a prior policy to guide the agent to a better starting position for exploration. The curriculum-based learning gradually reduces the agent's reliance on the demonstration, enabling a progressive transition from demonstration-guided to fully self-exploration. In this way, demonstrations directly guide RL exploration behavior and accelerate policy adaptation in a continual learning setting. Moreover, a self-evolving mechanism is used to continually store high-performance demonstrations into the repository during training.

Extensive experiments on continuous control tasks, including 2D navigation and MuJoCo locomotion, demonstrate that DGCRL consistently outperforms baseline methods by achieving the highest average performance, improved knowledge transfer, reduced forgetting, and training efficiency. We also note that the conventional forgetting metric is not suitable in demonstration-guided settings, as it can yield negative results. The sensitivity analysis indicates that a larger demonstration set generally leads to better performance. Furthermore, the ablation studies show that resetting both the actor and critic is important for stable and effective learning, and that combining a self-evolving demonstration repository with curriculum-based behavioral guidance is essential to DGCRL.

However, leveraging demonstrations to facilitate CRL remains in its early stages. Although our extensive evaluations on 2D Navigation and MuJoCo locomotion benchmarks confirm the effectiveness of DGCRL, we acknowledge that these are still simulation environments and extending DGCRL to real-world operational scenarios remains an important future direction. Particularly, DGCRL has high potential to suit real-world scenarios where exploration is expensive or safety-critical. By leveraging stored demonstrations, including suboptimal ones, it enables fast policy adaptation and improves the long-term reliability of RL agents in dynamic environments. Furthermore, in our experiments, the initial repository contains only 50-60 demonstrations. Although under the self-evolving strategy, new high-quality demonstrations were progressively added during training, the scale remained relatively small and no retrieval inefficiency was observed. Nevertheless, when scaling to larger demonstration repositories, retrieval efficiency may become a bottleneck. A promising direction for future work is to explore efficient demonstration management strategies, such as

indexing, labeling, pruning (e.g., clustering compression based on task similarity), or maintaining a fixed-size repository. Finally, conventional forgetting metrics have inherent limitations in demonstration-guided settings as negative scores may be produced, suggesting the need for more appropriate evaluation metrics.

**Broader Impact Statement**

This work studies continual reinforcement learning in dynamic environments and proposes a demonstration-guided framework to improve learning efficiency and knowledge reuse across tasks. By leveraging reusable demonstrations to guide exploration, the proposed method may contribute to the development of more adaptive and data-efficient autonomous systems. Potential application domains include robotics, autonomous navigation, energy management, and other sequential decision-making systems operating in non-stationary environments. In such settings, faster adaptation and reduced training cost may improve system reliability and reduce the need for extensive trial-and-error learning.

However, reinforcement learning systems deployed in real-world environments may also introduce risks. In safety-critical applications such as robotics or autonomous systems, incorrect demonstrations or poor-quality prior knowledge may bias the exploration process and lead to suboptimal or unsafe behaviors. Additionally, if demonstrations are derived from human data or operational systems, issues such as bias, privacy concerns, or unintended behavior transfer may arise.

To mitigate these risks, careful validation of demonstrations, monitoring of policy behavior, and safety constraints during deployment are important considerations. Furthermore, the experiments in this work are conducted in simulated environments, and additional evaluation in real-world settings is necessary before practical deployment. Future work will explore safety-aware demonstration selection and more robust mechanisms for managing large-scale demonstration repositories.

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

# Appendix

## A    Implementation Details

We provide the hyperparameters of DGCRL in Table 5. All experiments were implemented using PyTorch. The source code is available at: `https://github.com/XueYang0130/DGCRL.git`. For baseline implementation details, please refer to `https://github.com/HeyuanMingong/llirl`.

Table 5: DGCRL Hyperparameters and Their Values Across Environments

| Hyperparameter | Value & Description |
| --- | --- |
| $\alpha$ (Actor learning rate) | $3 \times 10^{-4}$ |
| $\beta$ (Critic learning rate) | $3 \times 10^{-4}$ |
| $\gamma$ (Discount factor) | 0.99 (navigation v1, v2, v3), 0.95 (hopper, ant, half-cheetah) |
| $\tau$ (Soft update rate) | 0.005 |
| Action Noise | 0.05 (navigation v1, v2, v3, hopper), 0.1 (ant, half-cheetah) |
| Policy Noise | 0.02 (navigation v1, v2, v3), 0.1 (hopper), 0.05 (ant, half-cheetah) |
| Actor Update Delay | 2 |
| Actor Architecture | [400, 300] |
| Critic Architecture | [400, 300] |
| H (Task Horizon) | 100 |
| Episodes | 100 (navigation v1, v2, v3), 200(hopper), 500 (ant, half-cheetah) |
| Roll-back Steps | 3 (navigation v1), 1 (navigation v2), 5 (navigation v3), 10 (hopper, ant, half-cheetah) |
| Demonstration Quantity | 50 (navigation v1, v2, v3, ant, hopper); 60 (half-cheetah) |

## B    Theoretical Analysis

We provide the theoretical analysis of the regret upper bound and the sampling complexity.

### B.1    Regret Upper Bound

In CRL settings with $N$ tasks, we represent the task sequence as a set of MDPs, i.e., $\{\mathcal{M}_1, \mathcal{M}_2, ..., \mathcal{M}_N\}$. We assume there is a demonstration set $\mathcal{D}_k$ for each $\mathcal{M}_k$. The policy that directly follows the demonstration is referred to as the guide policy $\pi_g$. We extend the theoretical framework proposed in Uchendu et al. (2023) to the setting.

**Assumption A.1** (Quality of guide-policy $\pi_g$ Uchendu et al. (2023)) Assume there exists a feature mapping function $\phi : \mathcal{S} \rightarrow \mathbb{R}^d$, that for any policy $\pi$, $Q^\pi(s, a)$ and $\pi(s)$ depends on $s$ only through $\phi(s)$. For each task $\mathcal{M}_k$, assume that there exists a guide policy $\pi_{g,k}$, such that for all states $s$ and time steps $h$ in task $M_k$, the following holds:

$$\sup_{s,h} \frac{d_h^{\pi^{*,k}}(\phi(s))}{d_h^{\pi_{g,k}}(\phi(s))} \leq C_k, \tag{11}$$

where $d_h^{\pi^{*,k}}(\phi(s))$ is the state feature distribution at time step $h$ for the optimal policy $\pi^*$ in task $\mathcal{M}_k$; $d_h^{\pi_{g,k}}(\phi(s))$ is the state feature distribution for the guide policy $\pi_{g,k}$; $C_k$ is the concentratability coefficient (Rashidinejad et al., 2021; Guo et al., 2023) in task $\mathcal{M}_k$, measures the degree to which the guide policy $\pi_{g,k}$ encompasses the optimal policy $\pi^{*,k}$ within the feature space. It ensures that the likelihood of the optimal policy encountering a specific feature $\phi(s)$ does not surpass $C_k$ times the likelihood of the guide policy encountering the same feature.

This is a standard assumption in regret analysis of demonstration-guided RLUchendu et al. (2023); Lu et al. (2024), that there exists a feature mapping $\phi(s)$ that characterizes the underlying structure of the MDP. This

assumption is existential and does not require explicit construction. In practice, policy networks implicitly learn a representation of states through the neurons, which can be viewed as a practical approximation of such a mapping. Although the mapping is not explicitly computed, the learned latent representations play the same role of shaping state distributions. In DGCRL, the coefficient is closely related to the quality and coverage of demonstrations in the repository. High-performance demonstrations typically visit informative or near-optimal regions of the state space, thus including state distributions closer to those of the optimal policy. The demonstration quality directly reduces the effective concentratability coefficient.

**Assumption A.2** (Performance of the Exploration Algorithm in CRL Uchendu et al. (2023)) In the context of contextual bandits, there exists an exploration algorithm, $ExplorationOracle_{CRL}$, that is suitable for non-stationary environments, such that in task $\mathcal{M}_k{}^3$, its regret upper bound is given by:

$$\text{Regret}_k(T_k) \leq f_k(T_k, R_k), \tag{12}$$

where $T_k$ is the number of training steps in task $\mathcal{M}_k$; $R_k$ is the reward range in task $\mathcal{M}_k$; $f_k$ is a function that describes the regret upper bound in task $\mathcal{M}_k$.

In CRL, we need to compute the cumulative regret across all tasks. Given the training budget $T$, where $T = \sum_{k=1}^{K} T_k$, the total regret is defined as:

$$\text{Total Regret}(T) = \sum_{k=1}^{K} \text{Regret}_k(T_k). \tag{13}$$

For a single task $\mathcal{M}_k$, the regret is:

$$\mathbb{E}_{s_{0,k} \sim p_{0,k}} \left[ V^{*,k}(s_0) - V^{\pi_k}(s_0) \right] = \sum_{h=0}^{H-1} \mathbb{E}_{s \sim d_h^{\pi^{*,k}}} \left[ Q_h^{\pi_k}(s, \pi_h^{*,k}(s)) - Q_h^{\pi_k}(s, \pi_h^k(s)) \right],$$

$\pi_k$ is the learned policy or explore policy for task $\mathcal{M}_k$; $V^{*,k}(s_0)$ and $V^{\pi_k}(s_0)$ are the value functions of the optimal policy and the learned policy, respectively; $d_h^{\pi^{*,k}}$ is the state distribution at step $h$ under the optimal policy for task $\mathcal{M}_k$; $p_{0,k}$ is the distribution of the initial state of the $k$ task.

By applying Assumption B.1, we have:

**Lemma 1.** *The performance difference between the optimal policy and the current learning policy on the states visited by the optimal policy does not exceed $C$ times the performance difference measured on the states visited by the guide policy.*

$$\mathbb{E}_{s \sim d_h^{\pi^{*,k}}} \left[ Q_h^{\pi_k}(s, \pi_h^{*,k}(s)) - Q_h^{\pi_k}(s, \pi_h^k(s)) \right]$$
$$\leq C_k \mathbb{E}_{s \sim d_h^{\pi^{g,k}}} \left[ Q_h^{\pi_k}(s, \pi_h^{*,k}(s)) - Q_h^{\pi_k}(s, \pi_h^k(s)) \right].$$

*Proof.* Let $g(s) = Q_h^{\pi_k}(s, \pi_h^{*,k}(s)) - Q_h^{\pi_k}(s, \pi_h^k(s))$

We first convert the expectation into an integral over state features. Assume $Q_h^{\pi_k}(s, a)$, policy $\pi_h^{*,k}(s)$ and $\pi_h^k(s)$ only depend on the feature representation of the state $s$, i.e. $\phi(s)$. Therefore, we can rewrite

$$g(s) = g(\phi(s)) = Q_h^{\pi_k}(\phi(s), \pi_h^{*,k}(\phi(s))) - Q_h^{\pi_k}(\phi(s), \pi_h^k(\phi(s)))$$

Furthermore, the expectation can be rewritten as:

$$\mathbb{E}_{s \sim d_h^{\pi^{*,k}}}[g(s)] = \mathbb{E}_{\phi(s) \sim d_h^{\pi^{*,k}}}[g(\phi(s))] = \int g(\phi(s)) d_h^{\pi^{*,k}}(\phi(s)) d\phi(s)$$

---

$^3$We simplify the CRL problem as a sequence of contextual bandits.

According to Assumption B.1, we have:

$$d_h^{\pi^{*,k}}(\phi(s)) \leq C_k d_h^{\pi^{g,k}}(\phi(s))$$

So we have:

$$\int f(\phi(s)) \, d_h^{\pi^{*,k}}(\phi(s)) \, d\phi(s) \leq C_k \int f(\phi(s)) \, d_h^{\pi^{g,k}}(\phi(s)) \, d\phi(s)$$
$$= C_k \mathbb{E}_{\phi(s) \sim d_h^{\pi^{g,k}}} \left[ g(\phi(s)) \right]$$

Therefore we have:

$$\mathbb{E}_{s \sim d_h^{\pi^{*,k}}} [g(s)] \leq C_k \mathbb{E}_{\phi(s) \sim d_h^{\pi^{g,k}}} [g(\phi(s))]$$

$$\text{i.e.} \quad \mathbb{E}_{s \sim d_h^{\pi^{*,k}}} \left[ Q_h^{\pi_k}(s, \pi_h^{*,k}(s)) - Q_h^{\pi_k}(s, \pi_h^k(s)) \right]$$
$$\leq C_k \mathbb{E}_{s \sim d_h^{\pi^{g,k}}} \left[ Q_h^{\pi_k}(s, \pi_h^{*,k}(s)) - Q_h^{\pi_k}(s, \pi_h^k(s)) \right]$$

$$\square$$

For simplicity, we assume the total number of training steps $T_k$ is evenly distributed across H time steps and the maximal sum reward is $R = H - h$ since the $h$ step. Extending Assumption B.1 to a $H$-horizon problem, for task $\mathcal{M}_i$, we have:

$$\mathbb{E}_{s \sim d_h^{\pi^{g,k}}} \left[ Q_h^{\pi_k}(s, \pi_h^{*,k}(s)) - Q_h^{\pi_k}(s, \pi_h^k(s)) \right] \leq f_k \left( \frac{T_k}{H}, H - h \right).$$

Now give the upper bound on value loss for a single task. Substituting this result into the performance difference lemma, we get:

$$\mathbb{E}_{s_0 \sim p_{0,k}} \left[ V^{*,k}(s_0) - V^{\pi_k}(s_0) \right] \leq C_k \sum_{h=0}^{H-1} f_k \left( \frac{T_k}{H}, H - h \right).$$

We can then calculate the total regret and value loss upper bound. The total regret upper bound is:

$$\text{Total Regret}(T) = \sum_{k=1}^{K} \mathbb{E}_{s_0 \sim p_{0_k}} \left[ V^{*,k}(s_0) - V^{\pi_k}(s_0) \right]$$
$$\leq \sum_{k=1}^{K} C_k \sum_{h=0}^{H-1} f_k \left( \frac{T_k}{H}, H - h \right)$$

If for each task $\mathcal{M}_k$, the reward range is $R_k \leq 1$, the coverage constant for the guide policy is $C_k = C$, assuming we use $\epsilon$-greedy strategy and the regret upper bound of the exploration algorithm Langford & Zhang (2007) is given by:

$$f_k \left( \frac{T_k}{H}, H - h \right) \leq C(H - h) \left( \frac{H}{T_k} \right)^{1/3} \tag{14}$$

Then the total regret upper bound is:

$$\text{Total Regret}(T) \le \sum_{k=1}^{K} \sum_{h=0}^{H-1} f_k \left( \frac{T_k}{H}, H - h \right) = C H^{1/3} T_k^{-1/3} \sum_{k=1}^{K} \sum_{h=0}^{H-1} (H - h) \tag{15}$$

With the inner summation being calculated as

$$\sum_{h=0}^{H-1} (H - h) = \frac{H(H+1)}{2}$$

and assuming we are assigning training budget $T$ evenly on each task, i.e. $T_k = \frac{T}{K}$, the total regret is then:

$$\text{Total Regret}(T) \le C H^{1/3} \left( \frac{K}{T} \right)^{1/3} H(H+1)K = C H^{4/3} (H+1) K^{4/3} T^{-1/3} \tag{16}$$

The regret bound contains a $K^{4/3}$ factor reflecting that the difficulty of continual learning grows sub-linearly with the number of tasks. Empirically, we observe that DGCRL maintains strong performance as the task sequence grows to 50 tasks. This trend is consistent with the theoretical prediction that demonstration-guided exploration mitigates compounding across tasks.

## B.2   Sample Complexity

We now compute the sample complexity required to learn a $\delta$-optimal model. To ensure that the average policy value loss per task is less than $\delta$, i.e.

$$\Delta V_{avg} = \frac{Total Regret(T)}{K} \le \delta$$

With Equation 16, we have:

$$\Delta V_{avg} == C H^{4/3} (H+1) K^{1/3} T^{-1/3} \le \delta$$

Solve for $T$, we get the sample complexity[4]:

$$T \ge \left( \frac{C H^{4/3} (H+1) K^{1/3}}{\delta} \right)^3 = \frac{C^3 H^4 (H+1)^3 K}{\delta^3}. \tag{17}$$

The sample complexity is proportional to the number of tasks $K$. The more tasks there are in the CRL scenario, the more training budget is required. $T$ is also proportional to $H^4$. This means the longer the horizon is, the more difficult the learning is. The $\frac{1}{\delta^3}$ means that the better the policy is the more training budget is needed.

It is important to note that DGCRL differs fundamentally in paradigm from prior approaches such MOLe (Nagabandi et al., 2018) and DaCoRL (Zhang et al., 2023). These methods are not formulated under a regret minimization framework and do not provide regret bounds in their original analyses, making a direct bound-to-bound comparison infeasible. Nevertheless, qualitatively, since these prior methods mainly rely on self-exploration, they may suffer higher regret during the initial learning phase. By contrast, DGCRL leverages demonstrations to constrain early exploration, which reduces cumulative regret. A rigorous theoretical comparison under a unified framework is left for future work.

---

[4]The possible minimal number of training steps $T$ to reach a specific standard of policy.

