# OpenReview forum: "Demonstration-Guided Continual Reinforcement Learning in Dynamic Environments"
_TMLR — Under review for TMLR_

### Review · Reviewer_WWtD · 2026-05-05

**Summary Of Contributions:**

The authors propose demonstration-guided continual reinforcement learning (DGCRL) as a method for leveraging past experience explicitly for guiding the exploration behavior of new tasks. In a continual RL setting where tasks arrive sequentially, the agent progressively builds a repository of demonstrations as trajectories executed in past tasks. The agent selects one of these demonstrations to execute in the current problem and then engages in standard RL exploration from the final state of execution of the demonstration. Once a trajectory combining the part demonstration and part exploration outperforms the pure demonstration behavior in the current task, the trajectory is stored in memory and the demonstration horizon is reduced. The submission contains experiments in 2D navigation and Mujoco environments.

######## Strengths ########
- The idea of using memory explicitly to guide new behavior, as opposed to merely as a means for reducing forgetting, is interesting and promising.
- The principle of measuring when the current policy is better than past demonstrations to automatically grow the repository of demonstrations is clever.

######## Weaknesses ########
- Various key details about the proposed method are not provided in sufficient detail.
    - This idea of "default envs" is concerning at first glance: how similar are they to the real ones? How will evaluation account for this extra training? These details are not provided in Sec 5 or 6.
    - Another concern: why are demos only actions? Are these not grounded in states at all? Just literally action sequences to execute open-loop?
    - How do we identify the demo with the highest return in a task? Run them?
    - "The policy $\pi_i$ of the agent comprises two components: ..." — This paragraph seems to suggest that the agent executes one trajectory composing the demonstration and the self-exploration policy, and then one trajectory of evaluation. This means that half the data is never used for learning?
    - Are the policies task-specific? I believe that they are, because the authors use the notation $\pi_{e,i}$, where $i$ stands for the task index. If so, any measure of average performance or forgetting is not meaningful as past parameters are not affected. Only forward transfer is meaningful.
- If past demonstrations are indeed executed open loop, as suggested by the fact that only action sequences are stored, then the method makes critical assumptions that are not stated and that significantly reduce the scope of the claims.
    - There seems to be two key assumptions here that are necessary for the proposed method, as understood by the reviewer (with open-loop action sequences as demonstrations): 1) the initial state is the same across all episodes and tasks, and 2) there transition function is deterministic. This makes it so that the agent can exactly follow the demonstrations open loop. These assumptions should be explicitly stated in the manuscript, at a minimum in Section 3, since they significantly reduce the scope where the method is applicable to a small subset of MDPs.
- One of the motivations behind the method is to avoid the need to explicitly determine task similarity, as needed by e.g. Nagabandi et al. Yet the proposed approach requires explicitly identifying which is the most relevant demonstration from the repository for the current task (through some undisclosed mechanism).
    - Because the authors state that their process identifies the trajectory that achieves the highest return on the current task, this seems to suggest that their approach requires executing each prior demonstration on the current environment. As the number of tasks grows, this requires a significant amount of environment interaction. It is not clear how the experimental setting accounts for this additional interaction time.

**Additional Comments:**

The following points are provided as feedback to hopefully help better shape the submitted manuscript, but will not impact my recommendation in a major way.


Intro
- "state visitation distribution is unchanged" — this is often w.r.t. a shared policy trained on many past tasks, which directly influences behavior based on past experience. I agree with the authors that having a more explicit mechanism to guide behavior is promising, but this claim should be softened.
- The use of the term "demonstration" seems to be incorrect, since these are replay trajectories, but the idea comes across

Sec 5
- Is the 2D navigation environment discretized? The depiction in Fig. 2 suggests a grid environment, but the action space description (velocity in $[-0.1, 0.1]$) is continuous.

Sec 6
- "during evaluation the agent can retrieve the most suitable demonstration to directly guide its behavior." — This is the first mention of using demonstrations to guide behavior *during evaluation*. Does this only happen if $h$ is never reduced down to 0? Is $h$ modified individually for each task or is it a shared parameter across tasks?

Typos/style/grammar
- Citation format should be consistent, with parentheses (e.g., via \citep) around citations that are not used as nouns.

**Audience:**

Yes

**Audience Explanation:**

If the findings are refined to their appropriate scope, it is possible that the continual RL community would find this paper of interest.

**Broader Impact Concerns:**

None.

**Claims And Evidence:**

No

**Claims Explanation:**

If the reviewer's understanding is correct, then the claims should be substantially revised to reflect the true scope of the contributions:
- The proposed method applies to *deterministic* MDPs only, where both the initial state and the transition dynamics are deterministic
- There is a significant cost in data requirements for: 1) executing past demonstrations to determine the optimal one for each task, and 2) evaluating the current exploration policy compared to the demonstration returns. It is not clear whether this cost is accounted for in the evaluations of Sec 6.
- Complete details of the proposed method must be provided.

**Requested Changes:**

Please see weaknesses and issues with the scope of the claims. All suggested revisions in those boxes are critical in the eyes of the reviewer.

---

### Review · Reviewer_A6q8 · 2026-05-08

**Summary Of Contributions:**

This paper proposes a novel framework named "Demonstration-Guided Continual Reinforcement Learning (DGCRL)" aimed at addressing the "stability-plasticity dilemma" in dynamic environments. Unlike traditional methods that implicitly encode prior knowledge in model parameters , DGCRL constructs a non-parametric external, self-evolving demonstration repository to explicitly store high-quality trajectory-level action sequences as demonstrations. When facing a new task, the agent dynamically selects the most relevant demonstration and, combined with the curriculum-based strategy of Jump-Start RL, guides the initial exploration process of the model.

Although the authors conducted extensive experiments on 2D navigation and MuJoCo locomotion tasks and claimed that DGCRL comprehensively outperforms CRL baselines (such as MAML, LLIRL, etc.) in terms of average performance, forward transfer, and the mitigation of forgetting, an in-depth review reveals fatal flaws in the methodological and theoretical rigor of this work. Specifically: its experimental initialization mechanism, which pre-loads a large amount of expert data, leads to fundamentally unfair competition with baseline algorithms; furthermore, its core theoretical derivation of the regret upper bound relies on mathematical assumptions that are bound to fail in dynamic sequential tasks, making its theoretical claims completely vacuous and invalid under the actual setting.

**Key Strengths**

Combining Jump-Start RL with an external memory repository is a highly practical and intuitive approach that can effectively mitigate performance drops in sequential tasks.

The method consistently outperforms baseline methods across multiple environments with varying non-stationarity.
The paper effectively isolates the contributions of each component, proving that exploration combined with a self-evolving demonstration repository is vastly superior to previous trajectory-level replay baselines.

**Key Weaknesses**

The framework is essentially a straightforward engineering concatenation of off-the-shelf Jump-Start RL (JSRL) and a standard demonstration repository. While intuitively effective, it merely ports JSRL—originally designed for single-task cold starts—into a multi-task setting. It lacks substantive breakthroughs at the underlying algorithmic level and entirely bypasses the true core bottlenecks faced by deep neural networks in continual reinforcement learning (e.g., catastrophic representational interference and the Loss of Plasticity induced by continuous updates). Given the aforementioned unfair empirical setups and theoretical flaws, this mere "system stitching" is insufficient to stand as the core innovation of an academic study.

The paper pre-loads 50-60 pre-trained demonstrations for DGCRL (Table 5), yet compares it against baseline methods (such as MAML, LLIRL) that learn from scratch. This asymmetric injection of massive prior knowledge deprives its claims of "superior forward transfer" and "average performance" of a fair foundation.

The "negative forgetting" phenomenon emphasized in the paper is misleading. DGCRL relies on a non-parametric external database (directly saving trajectories), while baseline methods are limited by weight interference in neural networks. Claiming that the external database "does not forget" and comparing it with neural networks on this basis is a fundamentally unfair cross-paradigm comparison.

The regret upper bound in the appendix heavily relies on Assumption A.1 (the concentratability coefficient Ck is bounded). However, in dynamic environments with abrupt goal changes, the state feature distributions of the old guide policy and the new optimal policy have zero overlap, inevitably causing Ck to approach infinity. This renders its theoretical bound completely invalid in the evaluated non-stationary environments.

The paper primarily evaluates the method on simple 2D navigation and basic MuJoCo locomotion tasks (e.g., Hopper, Ant). These tasks feature relatively low-dimensional state-action spaces, and their inter-task non-stationarity (e.g., merely changing goal positions or target velocities) is overly basic. This is insufficient to fully demonstrate the algorithm's capabilities in complex, real-world sequential scenarios. The lack of empirical validation on more challenging, higher-dimensional standard CRL benchmarks (such as the Continual World CW10 task sequence based on Meta-World) significantly weakens the overall persuasiveness of the experimental results.

**Audience:**

Yes

**Audience Explanation:**

Integrating Jump-Start RL into continuous control tasks under non-stationary environments is an inspiring and practical engineering pipeline. Researchers dedicated to robotic control, sim-to-real transfer, or applied lifelong learning might find the engineering combination of a self-evolving repository and curriculum-based exploration highly valuable for reference, despite the obvious flaws in the current paper's theoretical derivations and the fairness of its baseline evaluations.

**Broader Impact Concerns:**

The authors have adequately discussed the broader impacts in the dedicated "Broader Impact Statement" section. They accurately point out that in the real world and in safety-critical applications (such as robotics), relying on external demonstrations introduces risks if these demonstrations contain suboptimal, biased, or unsafe behaviors. They also acknowledge the limitations of currently testing only in simulated environments. The current statement is sufficient and does not require mandatory additional broader impact considerations.

**Claims And Evidence:**

No

**Claims Explanation:**

The empirical claims regarding superior performance and forward transfer are, to a large extent, an artifact of the "unfair experimental setup" rather than a triumph of the algorithm's own mechanisms. Granting DGCRL unconditional access to 50-60 pre-trained demonstrations provides it with a massive initial advantage, a privilege not enjoyed by the compared baselines. Unless it is ensured that the baseline methods have access to the exact same pre-training data (e.g., via pre-filling the replay buffer or offline pre-training), this massive performance gap cannot be scientifically attributed to the DGCRL architecture.

Furthermore, the theoretical claims are completely unsupported. In the theoretical proofs, the authors simplify the sequential MDP problem into contextual bandits and rely on a concentratability assumption (A.1) that is mathematically destined to fail when facing out-of-distribution task shifts (such as changing goal coordinates). The bound derived in contextual bandits with bounded Ck cannot mathematically prove the algorithm's performance in the ever-changing sequences of MuJoCo MDPs.

**Requested Changes:**

Given that the method is fundamentally a systemic integration of existing techniques (JSRL and trajectory-level replay), the authors MUST significantly tone down their overstated claims of "underlying algorithmic breakthroughs" in the introduction and conclusion. The authors should objectively reframe their contribution as an "engineering and system-level empirical pipeline" and candidly acknowledge in the discussion the limitations of this external-mounting architecture in solving fundamental internal issues such as catastrophic forgetting and Loss of Plasticity in parametric networks.

The authors must re-evaluate the baseline methods by providing them with the exact same initial knowledge as DGCRL. For example, use the exact same 50-60 trajectories from the initial repository of DGCRL to fill the replay buffers of the baseline methods (such as Adaptive, LLIRL), or allow them to undergo equivalent pre-training on these preceding environments.
The authors must completely remove the regret upper bound analysis in Appendix B, or explicitly prove mathematically: when the transition probabilities or reward functions change fundamentally between tasks, why does Assumption A.1 (bounded Ck) still hold? If the goal moves from (0,0) to (10,10), the previous guide policy will visit completely disjoint state distributions, which inevitably causes Ck to approach infinity.

In Section 6.3.2, the paper only compares with Initial Trajectory Replay (ITR) and Evolving Trajectory Replay (ETR) that lack policy optimization capabilities. This only proves that "a system with underlying RL learning capabilities is superior to static trajectory replay," but it fails to prove that the DGCRL architecture is superior to existing trajectory reuse algorithms. The authors must conduct a core empirical comparison of DGCRL against recent state-of-the-art (SOTA) continual reinforcement learning algorithms that truly possess learning capabilities and are also based on trajectories or behavioral cloning (e.g., ClonEx-SAC cited in the related work) to prove the substantive advancement of its guidance mechanism.

The authors must test DGCRL on more challenging standard continual reinforcement learning benchmarks (e.g., the Continual World CW10 task sequence). This is crucial for verifying whether the framework can still maintain its claimed "superior forward transfer" and "negative forgetting" advantages when dealing with high-dimensional complex robotic manipulation tasks and more drastic environmental dynamics.

---

> ### Author Response · Authors · 2026-05-27
> **Rebuttal 3**
>
> **Q5:** The paper primarily evaluates the method on simple 2D navigation and basic MuJoCo locomotion tasks (e.g., Hopper, Ant). These tasks feature relatively low-dimensional state-action spaces, and their inter-task non-stationarity (e.g., merely changing goal positions or target velocities) is overly basic. This is insufficient to fully demonstrate the algorithm's capabilities in complex, real-world sequential scenarios. The lack of empirical validation on more challenging, higher-dimensional standard CRL benchmarks (such as the Continual World CW10 task sequence based on Meta-World) significantly weakens the overall persuasiveness of the experimental results.
>
> **Rebuttal** : We agree that evaluating DGCRL on more challenging benchmarks such as Continual World would be a valuable future direction. At the same time, we would like to clarify that the current evaluation is designed to support the main goal of this paper: introducing and validating a demonstration-guided paradigm for continual RL, rather than providing an exhaustive benchmark study across all CRL domains.
> The environments used are not arbitrarily “toy” settings. They follow prior CRL/lifelong RL studies, especially LLIRL[1], which evaluates on the same 2D navigation and MuJoCo locomotion tasks. We use this to direct compare with prior CRL baselines. We have mentioned this in Section 5.
> Therefore, while we agree that Continual World would be an interesting extension, we do not believe it is necessary for establishing the core contribution of this paper. The current experiments are sufficient for the paper’s main claim: DGCRL provides a new behavior-level knowledge reuse framework for CRL and is effective on established dynamic-control benchmarks. We will revise the manuscript to more clearly position Continual World-style task as future work.
>
> **Q6:** The empirical claims regarding superior performance and forward transfer are, to a large extent, an artifact of the "unfair experimental setup" rather than a triumph of the algorithm's own mechanisms. Granting DGCRL unconditional access to 50-60 pre-trained demonstrations provides it with a massive initial advantage, a privilege not enjoyed by the compared baselines. Unless it is ensured that the baseline methods have access to the exact same pre-training data (e.g., via pre-filling the replay buffer or offline pre-training), this massive performance gap cannot be scientifically attributed to the DGCRL architecture.
>
> Furthermore, the theoretical claims are completely unsupported. In the theoretical proofs, the authors simplify the sequential MDP problem into contextual bandits and rely on a concentratability assumption (A.1) that is mathematically destined to fail when facing out-of-distribution task shifts (such as changing goal coordinates). The bound derived in contextual bandits with bounded Ck cannot mathematically prove the algorithm's performance in the ever-changing sequences of MuJoCo MDPs.
>
> **Rebuttal** We have addressed the empirical fairness concern above; here we focus on the theoretical concern raised in this part of the review.
>
> We would like to clarify that the theoretical analysis in Appendix B is based on the standard analysis used in demonstration-guided RL, particularly JSRL [2]. JSRL also relies on a guide-policy coverage/concentratability assumption and introduces a contextual-bandit-style exploration oracle to establish its sample-complexity bound. Our analysis extends this framework from the single-task JSRL setting to the continual setting by applying the guide-policy coverage assumption per task.
>
> Importantly, Assumption A.1 in our paper is not made with respect to an arbitrary old policy or a fixed demonstration from a previous task. It is stated per task: for each task $M_k$, there exists a guide policy $\pi_{g,k}$, whose feature-space visitation distribution covers that of the task-specific optimal policy up to coefficient Ck. This is aligned with the design of DGCRL, which dynamically selects the most relevant demonstration for each new task rather than blindly reusing a fixed old guide policy.
>
> We agree that the bound should not be interpreted as a guarantee under arbitrary out-of-distribution task shifts where no useful demonstration has feature-space overlap with the new optimal behavior. However, this makes the result conditional, not vacuous. Similar coverage assumptions are standard in demonstration-guided RL theory. We will revise Appendix B to state the scope more explicitly: the bound characterizes the regime in which the selected guide demonstration provides sufficient feature-space coverage of useful behavior in the current task.
>
> Rejecting our analysis solely because it relies on a bounded concentratability assumption would also reject the standard theoretical basis used by JSRL and related demonstration-guided RL analyses. The appropriate interpretation is that the result is conditional, not that it is mathematically meaningless.

---

### Review · Reviewer_1JnK · 2026-05-17

**Summary Of Contributions:**

The paper proposes a new method fo Continual Reinforcement Learning called Demonstration-Guided Continual Reinforcement Learning (DGCRL), which creates a demonstration repository where prior knowledge from the model is stored. The demonstrations in that repository are then used to guide RL exploration and adaptation, leading to superior average performance, enhanced knowledge transfer, mitigation of forgetting, and training efficiency.

**Audience:**

Yes

**Audience Explanation:**

Continual Learning in general, and Continural RL in specific are quite important fields with many practitioners who are part of the TMLR audience, and the results of the paper look quite groundbreaking, but without the extra details I'm not sure how useful it would be.

**Claims And Evidence:**

No

**Claims Explanation:**

While the results are amazing there are many unclear details about the algorithm and its computational cost that are fundamental for understanding how it works and for doing a fair comparison against the baselines.
First it's unclear how the demonstration set $\Pi_g$ is initialized and the impact of said initialization on the algorithm's performance. Furthermore the authors' describe one of the steps of the algorithm as "Select the demonstration πg,i that performs the best on task Mi", does that mean that we must run all of the demonstrations on the task? And how many times, given that RL environments can be stochastic? What's the added computational cost of doing that at the start of each new task compared to the cost of the other baselines?
Another important question that's not addressed is the impace of the size of $Pi_g$ for the performance of DGCRL.
Finally, it seems that in DGCRL the model can train until h goes to 0 for each task, which not only once again seems like it would add a significant computational cost compared to some of the baselines, but also means that the algorithm requires one to control when the task changes, which is not always the case in Continual Learning (where one can have the tasks changing at fixed intervals), pointing out 2 more important details that the paper glosses over.

**Requested Changes:**

More details on the initialization of $\Pi_g$ and its impact on the algorithm's performance, ablations on the size of $\Pi_g$, more details on how the $\pi_{g, i}$'s are evaluated and the computational cost of doing so in stochastic environemnts, computational cost comparison between the algorithms, not just number of steps, and an analysis of how the algorithm performs when the task changes automatically after K evaluations are all needed for the paper to be up to the expected standards of TLMR.
Nit: When the demonstration set $\Pi_g$ is defined you do not explain what the subscript g means, but you keep using it during the explanation.